# FAST UPDATING TRUNCATED SVD FOR REPRESENTATION LEARNING WITH SPARSE MATRICES

**Haoran Deng[1], Yang Yang[1]\*, Jiahe Li[1], Cheng Chen[2], Weihao Jiang[2], Shiliang Pu[2]**
[1]Zhejiang University, [2]Hikvision Research Institute
{denghaoran, yangya, jiaheli}@zju.edu.cn
{chencheng16, jiangweihao5, pushiliang.hri}@hikvision.com

## ABSTRACT

Updating a truncated Singular Value Decomposition (SVD) is crucial in representation learning. In practice, when dealing with evolving large-scale data matrices, aligning SVD-based models with fast-paced updates becomes critically important. Existing methods for updating truncated SVDs employ Rayleigh-Ritz projection procedures, where projection matrices are augmented based on original singular vectors. However, these methods are inefficient due to the densification of the update matrix and the application of the projection to all singular vectors. To overcome these limitations, we introduce a novel method for dynamically approximating the truncated SVD of a sparse and temporally evolving matrix. Our approach leverages sparsity in the orthogonalization process of augmented matrices and utilizes an extended decomposition to independently store projections in the column space of singular vectors. Numerical experiments demonstrate the efficiency of our method is improved by an order of magnitude compared to previous methods. Remarkably, this improvement is achieved while maintaining a comparable precision to existing approaches. Codes are available[1].

## 1 INTRODUCTION

Truncated Singular Value Decompositions (truncated SVDs) are widely used in various machine learning tasks, including computer vision (Turk & Pentland, 1991), natural language processing (Deerwester et al., 1990; Levy & Goldberg, 2014), recommender systems (Koren et al., 2009) and graph representation learning (Ramasamy & Madhow, 2015; Abu-El-Haija et al., 2021; Cai et al., 2022). Representation learning with a truncated SVD has several benefits, including the absence of gradients and hyperparameters, better interpretability derived from the optimal approximation properties, and efficient adaptation to large-scale data using randomized numerical linear algebra techniques (Halko et al., 2011; Ubaru et al., 2015; Musco & Musco, 2015).

However, large-scale data matrices frequently undergo temporal evolution in practical applications. Consequently, it is imperative for a representation learning system that relies on the truncated SVD of these matrices to adjust the representations based on the evolving data. Node representation for a graph, for instance, can be computed with a truncated SVD of the adjacency matrix, where each row in the decomposed matrix corresponds to the representation of a node (Ramasamy & Madhow, 2015). When the adjacency matrix undergoes modifications, it is necessary to update the corresponding representation (Zhu et al., 2018; Zhang et al., 2018; Deng et al., 2023). Similar implementations exist in recommender systems based on the truncated SVD of an evolving and sparse user-item rating matrix (Sarwar et al., 2002; Cremonesi et al., 2010; Du et al., 2015; Nikolakopoulos et al., 2019). This requirement for timely updates emphasizes the significance of keeping SVD-based models in alignment with the ever-evolving data.

Over the past few decades, methods recognized as the Rayleigh-Ritz projection process (Zha & Simon, 1999; Vecharynski & Saad, 2014; Yamazaki et al., 2015; Kalantzis et al., 2021) have been widely adopted for their high accuracy. Specifically, they construct the projection matrix by augmenting the columns of the current singular vector to an orthonormal matrix that roughly captures

---

\*Corresponding author
[1]https://github.com/zjunet/IncSVD.

the column space of the updated singular vector. Notably, the augmentation procedure typically thickens sparse areas of the matrices, rendering the inefficiency of these algorithms. Moreover, due to the requirement of applying the projection process to all singular vectors, these methods become impractical in situations involving frequent updates or using only a portion of the representations in the downstream tasks.

In this paper, we present a novel algorithm for fast updating truncated SVDs in sparse matrices.

**Contributions.**

1. We study the orthogonalization process of the *augmented matrix* performed in an inner product space isometric to the column space of the *augmented matrix*, which can take advantage of the sparsity of the updated matrix to reduce the time complexity. Besides, we propose an extended decomposition for the obtained orthogonal basis to efficiently update singular vectors.

2. We propose an algorithm for approximately updating the rank-$k$ truncated SVD with a theoretical guarantee (Theorem 1), which runs at the *update sparsity* time complexity when considering $k$ and $s$ (the rank of the updated matrix) as constants. We also propose two variants of the algorithm that can be applied to cases where $s$ is large.

3. We perform numerical experiments on updating truncated SVDs for sparse matrices in various real-world scenarios, such as representation learning applications in graphs and recommendations. The results demonstrate that the proposed method achieves a speed improvement of an order of magnitude compared to previous methods, while still maintaining a comparable accuracy.

## 2 BACKGROUND AND NOTATIONS

The singular value decomposition (SVD) of an $m$-by-$n$ real data matrix $\boldsymbol{A}$ is denoted by $\boldsymbol{A} = \boldsymbol{U}\boldsymbol{\Sigma}\boldsymbol{V}^\top$, where $\boldsymbol{U} \in \mathbb{R}^{m \times m}$, $\boldsymbol{V} \in \mathbb{R}^{n \times n}$ are orthogonal and $\boldsymbol{\Sigma} \in \mathbb{R}^{m \times n}$ is a rectangular diagonal with non-negative real numbers sorted across the diagonal. The rank-$k$ truncated SVD of $\mathbf{A}$ is obtained by keeping only the first $k$ largest singular values and using the corresponding $k$ columns of $\mathbf{U}$ and $\mathbf{V}$, which is denoted by $\mathbf{U}_k\boldsymbol{\Sigma}_k\mathbf{V}_k = \mathrm{SVD}_k(\mathbf{A})$.

In this paper, we consider the problem of updating the truncated SVD of a sparse matrix by adding new rows (or columns) and low-rank modifications of weights. Specifically, when a truncated SVD of data matrix $\mathbf{A} \in \mathbb{R}^{m \times n}$ is available, our goal is to approximate the truncated SVD of the new data matrix $\overline{\boldsymbol{A}}$ with addition of rows $\boldsymbol{E}_r \in \mathbb{R}^{s \times n}$ (or columns $\boldsymbol{E}_c \in \mathbb{R}^{m \times s}$) or low-rank modifications $\boldsymbol{D}_m \in \mathbb{R}^{m \times s}$, $\boldsymbol{E}_m \in \mathbb{R}^{s \times n}$ to $\boldsymbol{A}$.

$$\overline{\boldsymbol{A}} = \begin{bmatrix} \boldsymbol{A} \\ \boldsymbol{E}_r \end{bmatrix}, \quad \overline{\boldsymbol{A}} = [\boldsymbol{A} \ \boldsymbol{E_c}], \quad \text{or} \quad \overline{\boldsymbol{A}} = \boldsymbol{A} + \boldsymbol{D}_m\boldsymbol{E}_m^\top$$

In several related works, including Zha-Simon's, a key issue often revolves around the optimization of the QR decomposition of $(\boldsymbol{I} - \boldsymbol{U}_k\boldsymbol{U}_k^\top)\boldsymbol{B}$ matrix. Specifically, given an orthonormal matrix $\boldsymbol{U}_k$ and a sparse matrix $\boldsymbol{B}$, we refer to $(\boldsymbol{I} - \boldsymbol{U}_k\boldsymbol{U}_k^\top)\boldsymbol{B}$ as the **Augmented Matrix** with respect to $\boldsymbol{U}_k$, where its column space is orthogonal to $\boldsymbol{U}_k$.

**Notations.** In this paper, we use the lowercase letter $x$, bold lowercase letter $\boldsymbol{x}$ and bold uppercase letter $\boldsymbol{X}$ to denote scalars, vectors, and matrices, respectively. Moreover, $nnz(\boldsymbol{X})$ denotes the number of non-zero entries in a matrix, $\overline{\boldsymbol{X}}$ denotes the updated matrix, $\widetilde{\boldsymbol{X}}_k$ denotes the rank-$k$ approximation of a matrix, and $range(\boldsymbol{X})$ denotes the matrix's column space. A table of notations commonly used in this paper is listed in Appendix B.

### 2.1 RELATED WORK

**Projection Viewpoint.** Recent perspectives (Vecharynski & Saad, 2014; Kalantzis et al., 2021) frame the prevailing methodologies for updating the truncated SVD as instances of the Rayleigh-Ritz projection process, which can be characterized by following steps.

1. Augment the singular vector $\boldsymbol{U}_k$ and $\boldsymbol{V}_k$ by adding extra columns (or rows), resulting in $\widehat{\boldsymbol{U}}$ and $\widehat{\boldsymbol{V}}$, respectively. This augmentation is performed so that $range(\widehat{\boldsymbol{U}})$ and $range(\widehat{\boldsymbol{V}})$ can effectively approximate and capture the updated singular vectors' column space.

2. Compute $\boldsymbol{F}_k, \boldsymbol{\Theta}_k, \boldsymbol{G}_k = \mathrm{SVD}_k(\widehat{\boldsymbol{U}}^\top \boldsymbol{A} \widehat{\boldsymbol{V}})$.

3. Approximate the updated truncated SVD by $\widehat{\boldsymbol{U}} \boldsymbol{F}_k, \boldsymbol{\Theta}_k, \widehat{\boldsymbol{V}} \boldsymbol{G}_k$.

**Zha-Simon's Scheme.** For $\widetilde{\boldsymbol{A}}_k = \boldsymbol{U}_k \boldsymbol{\Sigma}_k \boldsymbol{V}_k^\top$, let $(\boldsymbol{I} - \boldsymbol{U}_k \boldsymbol{U}_k^\top) \boldsymbol{E}_c = \boldsymbol{QR}$, where $\boldsymbol{Q}$'s columns are orthonormal and $\boldsymbol{R}$ is upper trapezoidal. Zha & Simon (1999) method updates the truncate SVD after appending new columns $\boldsymbol{E}_c \in \mathbb{R}^{m \times s}$ to $\boldsymbol{A} \in \mathbb{R}^{m \times n}$ by

$$\overline{\boldsymbol{A}} \approx \begin{bmatrix} \widetilde{\boldsymbol{A}}_k & \boldsymbol{E}_c \end{bmatrix} = [\boldsymbol{U}_k \ \boldsymbol{Q}] \begin{bmatrix} \boldsymbol{\Sigma}_k & \boldsymbol{U}_k^\top \boldsymbol{E}_c \\ & \boldsymbol{R} \end{bmatrix} \begin{bmatrix} \boldsymbol{V}_k^\top & \\ & \boldsymbol{I} \end{bmatrix} = ([\boldsymbol{U}_k \ \boldsymbol{Q}] \boldsymbol{F}_k) \boldsymbol{\Theta}_k (\begin{bmatrix} \boldsymbol{V}_k & \\ & \boldsymbol{I} \end{bmatrix} \boldsymbol{G}_k)^\top \quad (1)$$

where $\boldsymbol{F}_k, \boldsymbol{\Theta}_k, \boldsymbol{G}_k = \mathrm{SVD}_k(\begin{bmatrix} \boldsymbol{\Sigma}_k & \boldsymbol{U}_k^\top \boldsymbol{E}_c \\ & \boldsymbol{R} \end{bmatrix})$. The updated approximate left singular vectors, singular values and right singular vectors are $[\boldsymbol{U}_k \ \boldsymbol{Q}] \boldsymbol{F}_k, \boldsymbol{\Theta}_k, \begin{bmatrix} \boldsymbol{V}_k & \\ & \boldsymbol{I} \end{bmatrix} \boldsymbol{G}_k$, respectively.

When it comes to weight update, let the QR-decomposition of the augmented matrices be $(\boldsymbol{I} - \boldsymbol{U}_k \boldsymbol{U}_k^\top) \boldsymbol{D}_m = \boldsymbol{Q}_D \boldsymbol{R}_D$ and $(\boldsymbol{I} - \boldsymbol{V}_k \boldsymbol{V}_k^\top) \boldsymbol{E}_m = \boldsymbol{Q}_E \boldsymbol{R}_E$, then the update procedure is

$$\overline{\boldsymbol{A}} \approx \widetilde{\boldsymbol{A}}_k + \boldsymbol{D}_m \boldsymbol{E}_m^\top = [\boldsymbol{U}_k \ \boldsymbol{Q}_D] (\begin{bmatrix} \boldsymbol{\Sigma}_k & \boldsymbol{0} \\ \boldsymbol{0} & \boldsymbol{0} \end{bmatrix} + \begin{bmatrix} \boldsymbol{U}_k^\top \boldsymbol{D}_m \\ \boldsymbol{R}_D \end{bmatrix} \begin{bmatrix} \boldsymbol{V}_k^\top \boldsymbol{E}_m \\ \boldsymbol{R}_E \end{bmatrix}^\top) [\boldsymbol{V}_k \ \boldsymbol{Q}_E]^\top$$
$$= ([\boldsymbol{U}_k \ \boldsymbol{Q}_D] \boldsymbol{F}_k) \boldsymbol{\Theta}_k ([\boldsymbol{V}_k \ \boldsymbol{Q}_E] \boldsymbol{G}_k)^\top \quad (2)$$

where $\boldsymbol{F}_k, \boldsymbol{\Theta}_k, \boldsymbol{G}_k = \mathrm{SVD}_k(\begin{bmatrix} \boldsymbol{\Sigma}_k & \boldsymbol{0} \\ \boldsymbol{0} & \boldsymbol{0} \end{bmatrix} + \begin{bmatrix} \boldsymbol{U}_k^\top \boldsymbol{D}_m \\ \boldsymbol{R}_D \end{bmatrix} \begin{bmatrix} \boldsymbol{V}_k^\top \boldsymbol{E}_m \\ \boldsymbol{R}_E \end{bmatrix}^\top)$. The updated approximate truncated SVD is $[\boldsymbol{U}_k \ \boldsymbol{Q}_D] \boldsymbol{F}_k, \boldsymbol{\Theta}_k, [\boldsymbol{V}_k \ \boldsymbol{Q}_E] \boldsymbol{G}_k$.

**Orthogonalization of Augmented Matrix.** The above QR decomposition of the augmented matrix and the consequent compact SVD is of high time complexity and occupies the majority of the total time when the granularity of the update is large (i.e., $s$ is large). To reduce the time complexity, a series of subsequent methods have been developed based on a faster approximation of the orthogonal basis of the augmented matrix. Vecharynski & Saad (2014) used Lanczos vectors conducted by a Golub-Kahan-Lanczos (GKL) bidiagonalization procedure to approximate the augmented matrix. Yamazaki et al. (2015) and Ubaru & Saad (2019) replaced the above GKL procedure with Randomized Power Iteration (RPI) and graph coarsening, respectively. Kalantzis et al. (2021) suggested determining only the left singular projection subspaces with the augmented matrix set as the identity matrix, and obtaining the right singular vectors by projection.

## 3 METHODOLOGY

We propose an algorithm for fast updating the truncated SVD based on the Rayleigh-Ritz projection paradigm. In Section 3.1, we present a procedure for orthogonalizing the augmented matrix that takes advantage of the sparsity of the updated matrix. This procedure is performed in an inner product space that is isometric to the augmented space. In Section 3.2, we demonstrate how to utilize the resulting orthogonal basis to update the truncated SVD. We also propose an extended decomposition technique to reduce complexity. In Section 3.3, we provide a detailed description of the proposed algorithm and summarize the main findings in the form of theorems. In Section 3.4, we evaluate the time complexity of our algorithm in relation to existing methods.

### 3.1 FASTER ORTHOGONALIZATION OF AUGMENTED MATRIX

In this section, we introduce an inner product space that is isomorphic to the column space of the augmented matrix, where each element is a tuple of a sparse vector and a low-dimensional vector. The orthogonalization process in this space can exploit sparsity and low dimension, and the resulting orthonormal basis is bijective to the orthonormal basis of the column space of the augmented matrix.

Previous methods perform QR decomposition of the augmented matrix with $\boldsymbol{QR} = (\boldsymbol{I} - \boldsymbol{U}_k \boldsymbol{U}_k^\top) \boldsymbol{B}$, to obtain the updated orthogonal matrix. The matrix $\boldsymbol{Q}$ derived from the aforementioned procedure is not only orthonormal, but its column space is also orthogonal to the column space of $\boldsymbol{U}_k$, implying that matrix $[\boldsymbol{U}_k \ \boldsymbol{Q}]$ is orthonormal.

Let $\boldsymbol{Z} = (\boldsymbol{I} - \boldsymbol{U}_k \boldsymbol{U}_k^\top)\boldsymbol{B} = \boldsymbol{B} - \boldsymbol{U}_k \boldsymbol{C}$ be the augmented matrix, then each column of $\boldsymbol{Z}$ can be written as a sparse $m$-dimensional matrix of column vectors minus a linear combination of the column vectors of the $m$-by-$k$ orthonormal matrix $\boldsymbol{U}_k$ i.e., $\boldsymbol{z}_i = \boldsymbol{b}_i - \boldsymbol{U}_k \boldsymbol{C}[i]$. We refer to the form of a sparse vector minus a linear combination of orthonormal vectors as SV-LCOV, and its definition is as follows:

**Definition 1** (SV-LCOV). *Let $\boldsymbol{U}_k \in \mathbb{R}^{m \times k}$ be an arbitrary matrix satisfying $\boldsymbol{U}_k^\top \boldsymbol{U}_k = \boldsymbol{I}$, and let $\boldsymbol{b} \in \mathbb{R}^m$ be a sparse vector. Then, the SV-LCOV form of the vector $\boldsymbol{z} = (\boldsymbol{I} - \boldsymbol{U}_k \boldsymbol{U}_k^\top)\boldsymbol{b}$ is a tuple denoted as $(\boldsymbol{b}, \boldsymbol{c})_{\boldsymbol{U}_k}$, where $\boldsymbol{c} = \boldsymbol{U}_k^\top \boldsymbol{b}$.*

Converting columns of an augmented matrix into SV-LCOV is efficient, because the computation of $\boldsymbol{U}_k^\top \boldsymbol{b}$ can be done by extracting the rows of $\boldsymbol{U}_k$ that correspond to the nonzero elements of $\boldsymbol{b}$, and then multiplying them by $\boldsymbol{b}$.

**Lemma 1.** *For an orthonormal matrix $\boldsymbol{U}_k \in \mathbb{R}^{m \times k}$ with $\boldsymbol{U}_k^\top \boldsymbol{U}_k = \boldsymbol{I}$, turning $(\boldsymbol{I} - \boldsymbol{U}_k \boldsymbol{U}_k^\top)\boldsymbol{b}$ with a sparse vector $\boldsymbol{b} \in \mathbb{R}^m$ into SV-LCOV can be done in time complexity of $O(k \cdot nnz(\boldsymbol{b}))$.*

Furthermore, we define the scalar multiplication, addition and inner product (i.e., dot product) of SV-LCOV and show in Lemma 2 that these operations can be computed with low time complexity when $\boldsymbol{b}$ is sparse.

**Lemma 2.** *For an orthonormal matrix $\boldsymbol{U}_k \in \mathbb{R}^{m \times k}$ with $\boldsymbol{U}_k^\top \boldsymbol{U}_k = \boldsymbol{I}$, the following operations of SV-LCOV can be done in the following time.*

- *Scalar multiplication: $\alpha(\boldsymbol{b}, \boldsymbol{c})_{\boldsymbol{U}_k} = (\alpha\boldsymbol{b}, \alpha\boldsymbol{c})_{\boldsymbol{U}_k}$ in $O(nnz(\boldsymbol{b}) + k)$ time.*
- *Addition: $(\boldsymbol{b}_1, \boldsymbol{c}_1)_{\boldsymbol{U}_k} + (\boldsymbol{b}_2, \boldsymbol{c}_2)_{\boldsymbol{U}_k} = (\boldsymbol{b}_1 + \boldsymbol{b}_2, \boldsymbol{c}_1 + \boldsymbol{c}_2)_{\boldsymbol{U}_k}$ in $O(nnz(\boldsymbol{b}_1 + \boldsymbol{b}_2) + k)$ time.*
- *Inner product: $\langle(\boldsymbol{b}_1, \boldsymbol{c}_1)_{\boldsymbol{U}_k}, (\boldsymbol{b}_2, \boldsymbol{c}_2)_{\boldsymbol{U}_k}\rangle = \langle\boldsymbol{b}_1, \boldsymbol{b}_2\rangle - \langle\boldsymbol{c}_1, \boldsymbol{c}_2\rangle$ in $O(nnz(\boldsymbol{b}_1 + \boldsymbol{b}_2) + k)$ time.*

With the scalar multiplication, addition and inner product operations of SV-LCOV, we can further study the inner product space of SV-LCOV.

**Lemma 3** (Isometry of SV-LCOV). *For an orthonormal matrix $\boldsymbol{U}_k \in \mathbb{R}^{m \times k}$ with $\boldsymbol{U}_k^\top \boldsymbol{U}_k = \boldsymbol{I}$, let $\boldsymbol{B} \in \mathbb{R}^{m \times s}$ be arbitrary sparse matrix with the columns of $\boldsymbol{B} = [\boldsymbol{b}_1, ..., \boldsymbol{b}_s]$, then the column space of $(\boldsymbol{I} - \boldsymbol{U}_k \boldsymbol{U}_k^\top)\boldsymbol{B}$ is **isometric** to the inner product space of their SV-LCOV.*

The *isometry* of an inner product space here is a transformation that preserves the inner product, thereby preserving the angles and lengths in the space. From Lemma 3, we can see that in SV-LCOV, since the dot product remains unchanged, the orthogonalization process of an augmented matrix can be transformed into an orthogonalization process in the inner product space.

As an example, the Modified Gram-Schmidt process (i.e. Algorithm 1) is commonly used to compute an orthonormal basis for a given set of vectors. It involves a series of orthogonalization and normalization steps to produce a set of mutually orthogonal vectors that span the same subspace as the original vectors. Numerically, the entire process consists of only three types of operations in Lemma 2, so we can replace them with SV-LCOV operations to obtain a more efficient method (i.e. Algorithm 2).

---

**Algorithm 1:** Modified Gram-Schmidt

**Input:** $\boldsymbol{E} \in \mathbb{R}^{n \times s}, \boldsymbol{U}_k \in \mathbb{R}^{n \times k}$
**Output:** $\boldsymbol{Q} \in \mathbb{R}^{n \times s}, \boldsymbol{R} \in \mathbb{R}^{s \times s}$
1  $\boldsymbol{Q} \leftarrow (\boldsymbol{I} - \boldsymbol{U}_k \boldsymbol{U}_k^\top)\boldsymbol{E}$;
2  **for** $i = 1$ **to** $s$ **do**
3      $\alpha \leftarrow \sqrt{\langle\boldsymbol{q}_i, \boldsymbol{q}_i\rangle}$;
4      $\boldsymbol{R}_{i,i} \leftarrow \alpha$;
5      $\boldsymbol{q}_i \leftarrow \boldsymbol{q}_i / \alpha$;
6      **for** $j = i + 1$ **to** $s$ **do**
7         $\beta \leftarrow \langle\boldsymbol{q}_i, \boldsymbol{q}_j\rangle$;
8         $\boldsymbol{R}_{i,j} \leftarrow \beta$;
9         $\boldsymbol{q}_j \leftarrow \boldsymbol{q}_j - \beta\boldsymbol{q}_i$;
10     **end**
11 **end**
12 **return** $\boldsymbol{Q} = [\boldsymbol{q}_1, ..., \boldsymbol{q}_s], \boldsymbol{R}$

---

**Algorithm 2:** SV-LCOV's QR process

**Input:** $\boldsymbol{E} \in \mathbb{R}^{m \times s}, \boldsymbol{U}_k \in \mathbb{R}^{m \times k}$
**Output:** $\boldsymbol{B} \in \mathbb{R}^{m \times s}, \boldsymbol{C} \in \mathbb{R}^{k \times s}, \boldsymbol{R} \in \mathbb{R}^{s \times s}$
1  $\boldsymbol{B} \leftarrow \boldsymbol{E}, \quad \boldsymbol{C} \leftarrow \boldsymbol{U}_k^\top \boldsymbol{E}$;
2  **for** $i = 1$ **to** $s$ **do**
3      $\alpha \leftarrow \sqrt{\langle\boldsymbol{b}_i, \boldsymbol{b}_i\rangle - \langle\boldsymbol{c}_i, \boldsymbol{c}_i\rangle}$;
4      $\boldsymbol{R}_{i,i} \leftarrow \alpha$;
5      $\boldsymbol{b}_i \leftarrow \boldsymbol{b}_i / \alpha, \quad \boldsymbol{c}_i \leftarrow \boldsymbol{c}_i / \alpha$;
6      **for** $j = i + 1$ **to** $s$ **do**
7         $\beta \leftarrow \langle\boldsymbol{b}_i, \boldsymbol{b}_j\rangle - \langle\boldsymbol{c}_i, \boldsymbol{c}_j\rangle$;
8         $\boldsymbol{R}_{i,j} \leftarrow \beta$;
9         $\boldsymbol{b}_j \leftarrow \boldsymbol{b}_j - \beta\boldsymbol{b}_i, \quad \boldsymbol{c}_j \leftarrow \boldsymbol{c}_j - \beta\boldsymbol{c}_i$;
10     **end**
11 **end**
12 **return** $\boldsymbol{Q} = [(\boldsymbol{b}_1, \boldsymbol{c}_1)_{\boldsymbol{U}_k}, ..., (\boldsymbol{b}_s, \boldsymbol{c}_s)_{\boldsymbol{U}_k}], \boldsymbol{R}$

---

**Lemma 4.** *Given an orthonormal matrix $U_k \in \mathbb{R}^{m \times k}$ satisfying $U_k^\top U_k = I$, there exists an algorithm that can obtain a orthonormal basis of a set of SV-LCOV $\{(b_1, c_1)_{U_k}, ..., (b_s, c_s)_{U_k}\}$ in $O((nnz(\sum_{i=1}^{s} b_i) + k)s^2)$ time.*

**Approximating the augmented matrix with SV-LCOV.** The Modified Gram-Schdmit process is less efficient when $s$ is large. To this end, Vecharynski & Saad (2014) and Yamazaki et al. (2015) approximated the orthogonal basis of the augmented matrix with Golub-Kahan-Lanczos bidiagonalization(GKL) procedure (Golub & Kahan, 1965) and Randomized Power Iteration (RPI) process. We find they consists of three operations in Lemma 2 and can be transformed into SV-LCOV to improve efficiency. Limited by space, we elaborate the proposed algorithm of SV-LCOV approximation to the augmented matrix in Appendix D.1 and Appendix D.2, respectively.

## 3.2 AN EXTENDED DECOMPOSITION TO REDUCING COMPLEXITY

**Low-dimensional matrix mutliplication and sparse matrix addition.** Suppose an orthonormal basis $(b_1, c_1)_{U_k}, ..., (b_s, c_s)_{U_k}$ of the augmented matrix in the SV-LCOV is obtained, according to Definition 1, this orthonormal basis corresponds to the matrix $B - U_k C$ where the $i$-th column of $B$ is $b_i$. Regarding the final step of the Rayleigh-Ritz process for updating the truncated SVD by adding columns, we have the update procedure for the left singular vectors:

$$\overline{U_k} \leftarrow [U_k \; Q] F_k = U_k F_k[: k] + Q F_k[k :]$$
$$= U_k F_k[: k] + (B - U_k C) F_k[k :] \qquad (3)$$
$$= U_k (F_k[: k] - C F_k[k :]) + B F_k[k :]$$

where $F_k[: k]$ denotes the submatrix consisting of the first $k$ rows of $F_k$, and $F_k[k :]$ denotes the submatrix consisting of rows starting from the $(k + 1)$-th rows of $F_k$.

Equation (3) shows that each update of the singular vectors $U_k$ consists of the following operations:

1. A low-dimensional matrix multiplication to $U_k$ by a $k$-by-$k$ matrix $(F_k[: k] - C F_k[k :])$.

2. A *sparse matrix addition* to $U_k$ by a sparse matrix $B F_k[k :]$ that has at most $nnz(B) \cdot k$ non-zero entries. While $B F_k[k :]$ may appear relatively dense in the context, considering it as a sparse matrix during the process of *sparse matrix addition* ensures asymptotic complexity.

**An extended decomposition for reducing complexity.** To reduce the complexity, we write the truncated SVD as a product of the following five matrices:

$$U'_{m \times k} \cdot U''_{k \times k} \cdot \Sigma_{k \times k} \cdot V''^\top_{k \times k} \cdot V'^\top_{n \times k} \qquad (4)$$

with orthonormal $U = U' \cdot U''$ and $V' \cdot V''$ (but not $V'$ or $V''$), that is, the singular vectors will be obtained by the product of the two matrices. Initially, $U''$ and $V''$ are set by the $k$-by-$k$ identity matrix, and $U'$, $V'$ are set as the left and right singular vectors. Similar additional decomposition was used in Brand (2006) for updating of SVD without any truncation and with much higher complexity. With the additional decomposition presented above, two operations can be performed to update the singular vector:

$$\overline{U''} \leftarrow U''(F_k[: k] - C F_k[k :])$$
$$\overline{U'} \leftarrow U' + B F_k[k :]\overline{U''}^{-1} \qquad (5)$$

which is a low-dimensional matrix multiplication and a sparse matrix addition. And the update of the right singular vectors is

$$\overline{V_k} \leftarrow \begin{bmatrix} V_k & \\ & I \end{bmatrix} G_k = \begin{bmatrix} V_k \\ 0 \end{bmatrix} G_k[: k] + \begin{bmatrix} 0 \\ I \end{bmatrix} G_k[k :] \qquad (6)$$

where $G_k[: k]$ denotes the submatrix consisting of the first $k$ rows of $G_k$ and $G_k[k :]$ denotes the submatrix cosisting of rows starting from the $(k + 1)$-th rows of $G_k$. Now this can be performed as

$$\overline{V''} \leftarrow V'' G_k[: k]$$
$$\text{Append matrix } G_k[k :]\overline{V''}^{-1} \text{ to } V' \qquad (7)$$

Above we have only shown the scenario of adding columns, but a similar approach can be used to add rows and modify weights. Such an extended decomposition reduces the time complexity of

updating the left and right singular vectors, allowing them to be deployed to the large-scale matrix with large $m$ and $n$. In practical applications, the aforementioned extended decomposition might introduce potential numerical issues when the condition number of the matrix is large, even though in most cases these matrices have relatively low condition numbers. One solution to this issue is to reset the $k$-by-$k$ matrix to the identity matrix.

## 3.3 MAIN RESULT

Algorithm 3 and Algorithm 4 are the pseudocodes of the proposed algorithm for adding columns and modifying weights, respectively.

---

**Algorithm 3:** Add columns

**Input:** $U_k(U', U''), \Sigma_k, V_k(V', V''), E_c$

1 Turn $(I - U_k U_k^\top)E_c$ into SV-LCOV and get $Q(B, C), R$ with Algorithm 2;

2 $F_k, \Theta_k, G_k \leftarrow \text{SVD}_k(\begin{bmatrix} \Sigma_k & U_k^\top E_c \\ & R \end{bmatrix})$;

3 $U'' \leftarrow U''(F_k[: k] - CF_k[k :])$;

4 $U' \leftarrow U' + BF_k[k :]U''^{-1}$;

5 $\Sigma_k \leftarrow \Theta_k$;

6 $V'' \leftarrow V''G_k[: k]$;

7 Append new columns $G[k :]V''^{-1}$ to $V'$;

---

**Algorithm 4:** Update weights

**Input:** $U_k(U', U''), \Sigma_k, V_k(V', V''), D_n, E_m$

1 Turn $(I - U_k U_k^\top)D$ into SV-LCOV and get $Q_D(B_D, C_D), R_D$ with Algorithm 2;

2 Turn $(I - V_k V_k^\top)E_m$ into SV-LCOV and get $Q_E(B_E, C_E), R_E$ with Algorithm 2;

3 $F_k, \Sigma_k, G_k \leftarrow$
$\text{SVD}_k(\begin{bmatrix} \Sigma_k & 0 \\ 0 & 0 \end{bmatrix} + \begin{bmatrix} U_k^\top D_m \\ R_D \end{bmatrix}\begin{bmatrix} V_k^\top E_m \\ R_E \end{bmatrix}^\top)$;

4 $U'' \leftarrow U''(F_k[: k] - C_D F_k[k :])$;

5 $U' \leftarrow U' + B_D F_k[k :]U''^{-1}$;

6 $\Sigma_k \leftarrow \Theta_k$;

7 $V'' \leftarrow V''(G_k[: k] - C_E G_k[k :])$;

8 $V' \leftarrow V' + B_E G_k[k :]U''^{-1}$;

---

The time complexity of the proposed algorithm in this paper is summarized in Theorem 1 and a detailed examination of the time complexity is provided in Appendix F.

**Theorem 1** (Main result). *There is a data structure for maintaining an approximate rank-$k$ truncated SVD of $A \in \mathbb{R}^{m \times n}$ with the following operations in the following time.*

- *$Add rows(E_r)$: Update $U_k, \Sigma_k, V_k \leftarrow SVD_k(\begin{bmatrix} \widetilde{A_k} \\ E_r \end{bmatrix})$ in $O(nnz(E_r)(s+k)^2 + (s+k)^3)$ time.*

- *$Add columns(E_c)$: Update $U_k, \Sigma_k, V_k \leftarrow SVD_k(\begin{bmatrix} \widetilde{A_k} & E_c \end{bmatrix})$ in $O(nnz(E_c)(s+k)^2 + (s+k)^3)$ time.*

- *$Update weights(D_m, E_m)$: Update $U_k, \Sigma_k, V_k \leftarrow SVD_k(\widetilde{A_k} + D_m E_m^\top)$ in $O(nnz(D_m + E_m)(s+k)^2 + (s+k)^3)$ time.*

- *$Query(i)$: Return $U_k[i]$ (or $V_k[i]$) and $\Sigma_k$ in $O(k^2)$ time.*

*where $\widetilde{A_k} = U_k \Sigma_k V_k^\top$ and $s$ is the rank of the update matrix.*

**Remark 1.** *Please note that we uses $\widetilde{A_k}$, the best rank-$k$ approximation of $A$, rather than the original $A$ matrix as the starting point of the updating process. Therefore, the update here may not obtain the updated best rank-$k$ approximation of the new matrix.*

The proposed method can theoretically produce the same output as Zha & Simon (1999) method with a much lower time complexity.

**The proposed variants with the approximate augmented space.** To address updates with coarser granularity (i.e., larger $s$), we propose two variants of the proposed algorithm based on approximate augmented spaces with GKL and RPI (see section 3.1) in SV-LCOV, denoted with the suffixes -GKL and -RPI, respectively. The proposed variants procude theoretically the same output as Vecharynski & Saad (2014) and Yamazaki et al. (2015), repectively. We elaborate the proposed variants in Appendix D.3.

Table 1: Time complexity comparing to previous methods

| Algorithm | Asymptotic complexity |
|---|---|
| Kalantzis et al. (2021) | $(m+n)k^2 + nnz(\boldsymbol{A})k + (k+s)k^2$ |
| Zha & Simon (1999) | $(m+n)k^2 + nks + ns^2 + (k+s)^3$ |
| Vecharynski & Saad (2014) | $(m+n)k^2 + nkl + nnz(\boldsymbol{E})(k+l) + (k+s)(k+l)^2$ |
| Yamazaki et al. (2015) | $(m+n)k^2 + t(nkl + nl^2 + nnz(\boldsymbol{E})l) + (k+s)(k+l)^2$ |
| ours | $nnz(\boldsymbol{E})(k+s)^2 + (k+s)^3$ |
| ours-GKL | $nnz(\boldsymbol{E})(k^2 + sl + kl) + (k+s)(k+l)^2$ |
| ours-RPI | $nnz(\boldsymbol{E})(sl + l^2)t + nnz(\boldsymbol{E})k^2 + (k+s)(k+l)^2$ |

## 3.4 TIME COMPLEXITY COMPARING TO PREVIOUS METHODS

Table. 1 presents the comparison of the time complexity of our proposed algorithms with the previous algorithms when updating rows $\boldsymbol{E} \in \mathbb{R}^{s \times n}$. To simplify the results, we have assumed $s < n$ and omitted the big-$O$ notation. $l$ denotes the rank of approximation in GKL and RPI. $t$ denotes the number of iteration RPI performed.

Our method achieves a time complexity of $O(nnz(\boldsymbol{E}))$ for updating, without any $O(n)$ or $O(m)$ terms when $s$ and $k$ are considered as constants (i.e., the proposed algorithm is at the *update sparsity time complexity*). This is because SV-LCOV is used to obtain the orthogonal basis, eliminating the $O(n)$ or $O(m)$ terms when processing the augmented matrix. Additionally, our extended decomposition avoids the $O(n)$ or $O(m)$ terms when restoring the SV-LCOV and eliminates the projection in all rows of singular vectors. Despite the time complexity of the query increases from $O(k)$ to $O(k^2)$, this trade-off remains acceptable considering the optimizations mentioned above.

For updates with one coarse granularity (i.e. larger $s$), the proposed method of approximating the augmented space with GKL or RPI in the SV-LCOV space eliminates the squared term of $s$ in the time complexity, making the proposed algorithm also applicable to coarse granularity update.

## 4 NUMERICAL EXPERIMENT

In this section, we conduct experimental evaluations of the update process for the truncated SVD on sparse matrices. Subsequently, we assess the performance of the proposed method by applying it to two downstream tasks: (1) link prediction, where we utilize node representations learned from an evolving adjacency matrix, and (2) collaborative filtering, where we utilize user/item representations learned from an evolving user-item matrix.

### 4.1 EXPERIMENTAL DESCRIPTION

**Baselines**. We evaluate the proposed algorithm and its variants against existing methods, including Zha & Simon (1999), Vecharynski & Saad (2014) and Yamazaki et al. (2015). Throughout the experiments, we set $l$, the spatial dimension of the approximation, to 10 based on previous settings (Vecharynski & Saad, 2014). The required number of iterations for the RPI, denoted by $t$, is set to 3. In the methods of Vecharynski & Saad (2014) and Yamazaki et al. (2015), there may be differences in running time between 1) directly constructing the augmented matrix, and 2) accessing the matrix-vector multiplication as needed without directly constructing the augmented matrix. We conducted tests under both implementations and considered the minimum value as the running time.

**Tasks and Settings**. For the adjacency matrix, we initialize the SVD with the first 50% of the rows and columns, and for the user-item matrix, we initialize the SVD with the first 50% of the columns.

The experiment involves $\phi$ batch updates, adding $n/\phi$ rows and columns each time for a $2n$-sized adjacency matrix. For a user-item matrix with $2n$ columns, $n/\phi$ columns are added per update.

- **Link Prediction** aims at predicting whether there is a link between a given node pair. Specifically, each node's representation is obtained by a truncated SVD of the adjacency matrix. During the inference stage, we first query the representation obtained by the truncated SVD of the node pair

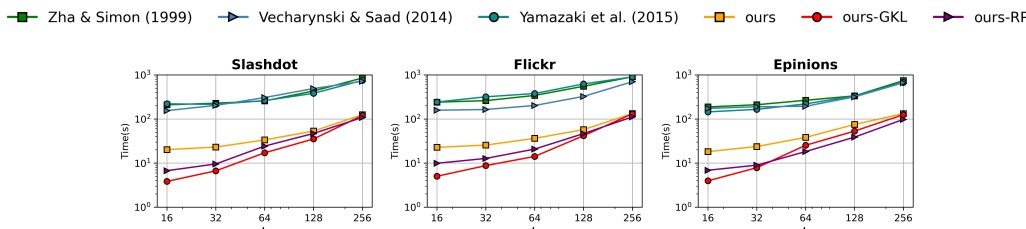

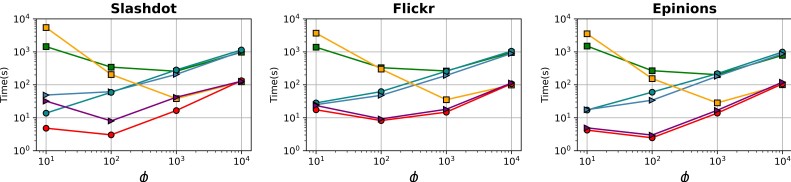

Figure 1: Computational efficiency of adjacency matrix when $k$ is $16, 32, 64, 128, 256$

Figure 2: Computational efficiency of adjacency matrix when $\phi$ is $10^1, 10^2, 10^3, 10^4$

$(i, j)$. A score is then performed, representing the inner product between pairs of nodes

$$\boldsymbol{U}[i]^\top \Sigma \boldsymbol{V}[j]$$

to make predictions. Maximum value in both directions is taken for undirected graphs. We sort the scores in the test set and label the edges between node pairs with high scores as positive ones.

Based on previous research, we create the training set $\mathcal{G}'$ by randomly removing 30% of the edges from the original graph $\mathcal{G}$. The node pairs connected by the eliminated edges are then chosen, together with an equal number of unconnected node pairs from $\mathcal{G}$, to create the testing set $\mathcal{E}_{test}$.

- **Collaborative Filtering** in recommender systems is a technique using a small sample of user preferences to predict likes and dislikes for a wide range of products. In this paper, we focus on predicting the values of the normalized user-item matrix.

  In this task, a truncated SVD is used to learn the representation for each user and item. And the value is predicted by the inner product between the representation of $i$-th user and $j$-th item with

$$\boldsymbol{U}[i]^\top \Sigma \boldsymbol{V}[j]$$

  The matrix is normalized by subtracting the average value of each item. Values in the matrix are initially divided into the training and testing set with a ratio of $8 : 2$.

**Datasets**. The link prediction experiments are conducted on three publicly available graph datasets, namely Slashdot (Leskovec et al., 2009) with $82, 168$ nodes and $870, 161$ edges, Flickr (Tang & Liu, 2009) with $80, 513$ nodes and $11, 799, 764$ edges, and Epinions (Richardson et al., 2003) with $75, 879$ nodes and $1, 017, 674$ edges. The social network consists of nodes representing users, and edges indicating social relationships between them. In our setup, all graphs are undirected.

For the collaborative filtering task, we use data from MovieLens (Harper & Konstan, 2015). The MovieLens25M dataset contains more than $2, 500, 000$ ratings for $62, 423$ movies. According to the selection mechanism of the dataset, all selected users rated at least 20 movies, ensuring the dataset's validity and a moderate level of density. All ratings in the dataset are integers ranging from 1 to 5.

## 4.2 EFFICIENCY STUDY

To study the efficiency of the proposed algorithm, we evaluate the proposed method and our optimized GKL and RPI methods in the context of link prediction and collaborative filtering tasks.

Experiments are conducted on the undirected graphs of Slashdot (Leskovec et al., 2009), Flickr (Tang & Liu, 2009), and Epinions (Richardson et al., 2003) to investigate link prediction. The obtained results are presented in Table 2, Fig. 1 and Fig. 2. Our examination metrics for the Efficiency section include runtime and **Average Precision (AP)** which is the percentage of correctly predicted edges in the predicted categories to the total number of predicted edges. Besides, we report the **Frobenius norm** between the $\boldsymbol{U}_k \boldsymbol{\Sigma}_k \boldsymbol{V}_k^\top$ and the train matrix. The results demonstrate

Table 2: Experimental results on adjacency matrix

| Method | Slashdot | | Flickr | | Epinions | |
|---|---|---|---|---|---|---|
| | Norm | AP | Norm | AP | Norm | AP |
| Zha & Simon (1999) | 792.11 | 93.40% | 2079.23 | 95.16% | 1370.26 | 95.62% |
| Vecharynski & Saad (2014) | 792.01 | 93.56% | 2079.63 | 95.11% | 1370.64 | 95.70% |
| Yamazaki et al. (2015) | 792.11 | 93.52% | 2079.28 | 95.14% | 1370.29 | 95.61% |
| ours | 792.11 | 93.41% | 2079.23 | 95.16% | 1370.26 | 95.62% |
| ours-GKL | 792.01 | 93.56% | 2079.63 | 95.11% | 1370.64 | 95.70% |
| ours-RPI | 792.11 | 93.50% | 2079.28 | 95.14% | 1370.29 | 95.61% |

Table 3: Experimental results of user-item matrix

| | Method | Batch Update | | Streaming Update | |
|---|---|---|---|---|---|
| | | Runtime | MSE | Runtime | MSE |
| $k$=16 | Zha & Simon (1999) | 192s | 0.8616 | 626s | 0.8616 |
| | Vecharynski & Saad (2014) | 323s | 0.8646 | 2529s | 0.8647 |
| | Yamazaki et al. (2015) | 278s | 0.8618 | 352s | 0.8619 |
| | ours | 23s | 0.8616 | 35s | 0.8616 |
| | ours-GKL | 18s | 0.8646 | 48s | 0.8647 |
| | ours-RPI | 45s | 0.8618 | 43s | 0.8619 |
| $k$=64 | Zha & Simon (1999) | 343s | 0.8526 | 2410s | 0.8527 |
| | Vecharynski & Saad (2014) | 124s | 0.8572 | 3786s | 0.8568 |
| | Yamazaki et al. (2015) | 313s | 0.8527 | 758s | 0.8528 |
| | ours | 49s | 0.8526 | 135s | 0.8527 |
| | ours-GKL | 45s | 0.8572 | 147s | 0.8568 |
| | ours-RPI | 98s | 0.8527 | 141s | 0.8528 |

that across multiple datasets, the proposed method and its variants have demonstrated significant increases in efficiency without compromising AP, reducing time consumption by more than $85\%$.

Table 3 demonstrates the results of four experiments conducted for the collaborative filtering task: batch update($\phi = 2000$) and streaming update($\phi = \#entries$) with $k = 16$ and $k = 64$. Our evaluation metrics include runtime and mean squared error (MSE). The results show that our method significantly reduces runtime while maintaining the comparable accuracy. It is difficult for existing methods to update large and sparse datasets, especially streaming updates, making real-time updates challenging. The methodology employed in our study effectively decreases the runtime by a factor of 10 or more.

## 4.3 VARYING $k$ AND $\phi$

We conduct link prediction experiments on three datasets for $k$ and $\phi$, aiming to explore the selection of variants and approaches in various scenarios. The results in Fig. 1 show that our optimized methods are significantly faster than the original ones for different choices of $k$. The performance of all methods slows as $k$ increases, which is consistent with the asymptotic complexity in Table 1.

We experimentally evaluate the performance of each method for different batch sizes $\phi$. As shown in Fig. 2, our methods (ours-GKL and ours-RPI) outperform others when a large number of entries are updated simultaneously. Experimental results show that the efficiency of approximating the augmented space improves significantly when $s$ is larger. Therefore, choosing the proposed variants (GKL or RPI) for larger $s$ is recommended.

## 5 CONCLUSION

In conclusion, we introduce a novel algorithm along with two variants for updating truncated SVDs with sparse matrices. Numerical experiments show a substantial speed boost of our method compared to previous approaches, while maintaining the comparable accuracy.

ACKNOWLEDGEMENT

This work is supported by Zhejiang NSF (LR22F020005).

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

## A    REPRODUCIBILITY

We released the implement of all the algorithms involved in the experiment as a python package. The code is available at: https://github.com/zjunet/IncSVD. We also make public our experimental code, which includes python implementations of all methods involved in the experiment, as well as the datasets. The code is available at https://github.com/HaoranDeng/IncSVDforICLR2024.

## B    NOTATIONS

Frequently used notations throughout the paper are summarized in Table 4.

Table 4: Frequently used notations

| Notation | Description |
|---|---|
| $\boldsymbol{A}$ | The data matrix |
| $\boldsymbol{E}_c$ | The update matrix (new columns) |
| $\boldsymbol{E}_r$ | The update matrix (new rows) |
| $\boldsymbol{D}_m, \boldsymbol{E}_m$ | The update matrix (low rank update of weight) |
| $\boldsymbol{I}$ | The identity matrix |
| $\boldsymbol{U}$ | The left singular vectors |
| $\boldsymbol{\Sigma}$ | The singular values |
| $\boldsymbol{V}$ | The right singular vectors |
| $(\boldsymbol{I} - \boldsymbol{U}_k\boldsymbol{U}_k)\boldsymbol{B}$ | The (left) augmented matrix |
| $(\boldsymbol{I} - \boldsymbol{V}_k\boldsymbol{V}_k)\boldsymbol{B}$ | The (right) augmented matrix |
| $m, n$ | The number of rows and columns of data matrix |
| $k$ | The rank of truncated SVD |
| $s$ | The rank of update matrix |
| $l$ | The rank of approximate augmented space |
| $t$ | The number of Random Power Iteration performed |
| $\boldsymbol{X}_k$ | A matrix with rank $k$ |
| $\boldsymbol{X}^\top$ | The transpose of matrix $\boldsymbol{X}$ |
| $\widetilde{\boldsymbol{X}_k}$ | A rank-$k$ approximation of $\boldsymbol{X}$ |
| $\overline{\boldsymbol{X}}$ | The updated matrix of $\boldsymbol{X}$ |
| $\|\boldsymbol{x}\|$ | $l_2$-norm of $\boldsymbol{x}$ |
| $\langle\boldsymbol{x}_1, \boldsymbol{x}_2\rangle$ | Dot product bewteen $\boldsymbol{x}_1, \boldsymbol{x}_2$ |
| $\mathrm{SVD}_k(\boldsymbol{X})$ | A rank-$k$ truncated SVD of $\boldsymbol{X}$ |

## C    OMITTED PROOFS

**Proof of Lemma 2**

*Proof.* We prove each of the three operations as follows.

- **Scalar multiplication.** It takes $O(nnz(b))$ time to get $\alpha\boldsymbol{b}$ and $O(k)$ time to get $\alpha\boldsymbol{c}$, respectively. Therefore the overall time complexity for scalar multiplication is $O(nnz(\boldsymbol{b}) + k)$.

- **Addition.** It takes $O(nnz(\boldsymbol{b}_1) + nnz(\boldsymbol{b}_2)) = O(nnz(\boldsymbol{b}_1 + \boldsymbol{b}_2))$ time to get $\boldsymbol{b}_1 + \boldsymbol{b}_2$ and $O(k)$ time to get $\boldsymbol{c}_1 + \boldsymbol{c}_2$, respectively. Therefore the overall time complexity for addition is $O(nnz(\boldsymbol{b}_1 + \boldsymbol{b}_2) + k)$.

- **Inner product.** It takes $O(nnz(\boldsymbol{b}_1) + nnz(\boldsymbol{b}_2)) = O(nnz(\boldsymbol{b}_1 + \boldsymbol{b}_2))$ time to get $\langle\boldsymbol{b}_1, \boldsymbol{b}_2\rangle$ and $O(k)$ time to get $\langle\boldsymbol{c}_1, \boldsymbol{c}_2\rangle$, respectively. Therefore the overall time complexity for inner product is $O(nnz(\boldsymbol{b}_1 + \boldsymbol{b}_2) + k)$.

$\square$

**Proof of Lemma 3**

*Proof.* Each of the three operations of SV-LCOV can correspond to the original space as follows.

- **Scalar multiplication.**

$$\alpha(\boldsymbol{b}, \boldsymbol{c})_{\boldsymbol{U}_k} = (\alpha \boldsymbol{b}, \alpha \boldsymbol{c})_{\boldsymbol{U}_k} = (\alpha \boldsymbol{b}) - \boldsymbol{U}_k(\boldsymbol{U}_k^\top \alpha \boldsymbol{b}) = \alpha(\boldsymbol{I} - \boldsymbol{U}_k \boldsymbol{U}_k^\top)\boldsymbol{b} \tag{8}$$

- **Addition.**

$$\begin{aligned}
(\boldsymbol{b}_1, \boldsymbol{c}_1)_{\boldsymbol{U}_k} + (\boldsymbol{b}_2, \boldsymbol{c}_2)_{\boldsymbol{U}_k} &= (\boldsymbol{b}_1 + \boldsymbol{b}_2, \boldsymbol{c}_1 + \boldsymbol{c}_2)_{\boldsymbol{U}_k} \\
&= (\boldsymbol{b}_1 + \boldsymbol{b}_2) - \boldsymbol{U}_k \boldsymbol{U}_k^\top(\boldsymbol{c}_1 + \boldsymbol{c}_2) \\
&= \boldsymbol{b}_1 - \boldsymbol{U}_k \boldsymbol{U}_k^\top \boldsymbol{c}_1 + \boldsymbol{b}_2 - \boldsymbol{U}_k \boldsymbol{U}_k^\top \boldsymbol{c}_2 \\
&= (\boldsymbol{I} - \boldsymbol{U}_k \boldsymbol{U}_k^\top)\boldsymbol{b}_1 + (\boldsymbol{I} - \boldsymbol{U}_k \boldsymbol{U}_k^\top)\boldsymbol{b}_2
\end{aligned} \tag{9}$$

- **Inner product.**

$$\begin{aligned}
\langle(\boldsymbol{b}_1, \boldsymbol{c}_1)_{\boldsymbol{U}_k}, (\boldsymbol{b}_2, \boldsymbol{c}_2)_{\boldsymbol{U}_k}\rangle &= \langle \boldsymbol{b}_1, \boldsymbol{b}_2 \rangle - \langle \boldsymbol{c}_1, \boldsymbol{c}_2 \rangle \\
&= \boldsymbol{b}_1^\top \boldsymbol{b}_2 - \boldsymbol{c}_1^\top \boldsymbol{c}_2 \\
&= \boldsymbol{b}_1^\top \boldsymbol{b}_2 - \boldsymbol{b}_1^\top \boldsymbol{U}_k \boldsymbol{U}_k^\top \boldsymbol{b}_2 \\
&= \boldsymbol{b}_1^\top \boldsymbol{b}_2 - 2\boldsymbol{b}_1^\top \boldsymbol{U}_k \boldsymbol{U}_k^\top \boldsymbol{b}_2 + \boldsymbol{b}_1^\top \boldsymbol{U}_k \boldsymbol{U}_k^\top \boldsymbol{b}_2 \\
&= \boldsymbol{b}_1^\top \boldsymbol{b}_2 - 2\boldsymbol{b}_1^\top \boldsymbol{U}_k \boldsymbol{U}_k^\top \boldsymbol{b}_2 + \boldsymbol{b}_1^\top \boldsymbol{U}_k \boldsymbol{U}_k^\top \boldsymbol{U}_k \boldsymbol{U}_k^\top \boldsymbol{b}_2 \\
&= (\boldsymbol{b}_1^\top - \boldsymbol{b}_1^\top \boldsymbol{U}_k \boldsymbol{U}_k^\top)(\boldsymbol{b}_2 - \boldsymbol{U}_k \boldsymbol{U}_k^\top \boldsymbol{b}_2) \\
&= ((\boldsymbol{I} - \boldsymbol{U}_k \boldsymbol{U}_k^\top)\boldsymbol{b}_1)^\top((\boldsymbol{I} - \boldsymbol{U}_k \boldsymbol{U}_k^\top)\boldsymbol{b}_2) \\
&= \langle(\boldsymbol{I} - \boldsymbol{U}_k \boldsymbol{U}_k^\top)\boldsymbol{b}_1, (\boldsymbol{I} - \boldsymbol{U}_k \boldsymbol{U}_k^\top)\boldsymbol{b}_2\rangle
\end{aligned} \tag{10}$$

□

**Proof of Lemma 4**

*Proof.* An orthogonal basis can be obtained by Algorithm 2, and we next analyze the time complexity of Algorithm 2.

Because the number of non-zero entries of any linear combination of a set of sparse vectors $\{\boldsymbol{b}_1, \boldsymbol{b}_2, ..., \boldsymbol{b}_k\}$ with size $k$ is at most $\sum_{i=1}^k nnz(\boldsymbol{b}_i)$, the number of non-zero entries of the sparse vector $(\boldsymbol{b})$ in each SV-LCOV during the process of Algorithm 2 are at most $nnz(\sum_{i=1}^k \boldsymbol{b}_i)$.

There are a total of $O(s^2)$ projections and subtractions of SV-LCOV in Algorithm 2, and by Lemma 2, the overall time complexity is $O((nnz(\sum_{i=1}^k \boldsymbol{b}_i) + k)s^2)$.

□

**Proof of Theorem 1**

*Proof.* The output of proposed method is theoretically equivalent to the output of Zha & Simon (1999), and the detailed time complexity analysis is given in Appendix F. □

## D  APPROXIMATING THE AUGMENTED SPACE

### D.1  APPROXIMATING THE AUGMENTED SPACE VIA GOLUB-KAHAN-LANCZOS PROCESS

| **Algorithm 5:** GKL | **Algorithm 6:** GKL with SV-LCOV |
|---|---|
| **Input:** $\boldsymbol{E} \in \mathbb{R}^{m \times s}, \boldsymbol{U}_k \in \mathbb{R}^{m \times k}, l \in \mathbb{N}^+$ | **Input:** $\boldsymbol{E} \in \mathbb{R}^{m \times s}, \boldsymbol{U}_k \in \mathbb{R}^{m \times k}, l \in \mathbb{N}^+$ |
| **Output:** $\boldsymbol{Q} \in \mathbb{R}^{n \times l}, \boldsymbol{P} \in \mathbb{R}^{s \times l}$ | **Output:** $\boldsymbol{B} \in \mathbb{R}^{n \times l}, \boldsymbol{C} \in \mathbb{R}^{s \times l}, \boldsymbol{P} \in \mathbb{R}^{s \times l}$ |
| 1  $\boldsymbol{Z} \leftarrow (\boldsymbol{I} - \boldsymbol{U}_k \boldsymbol{U}_k^\top)\mathbf{E}$; | 1  $\boldsymbol{B} \leftarrow \boldsymbol{E}, \quad \boldsymbol{C} \leftarrow \boldsymbol{U}_k^\top \boldsymbol{E}$; |
| 2  Choose $\boldsymbol{p}_1 \in \mathbb{R}^s, \|\boldsymbol{p}_1\| = 1$. Set $\beta_0 = 0$; | 2  Choose $\boldsymbol{p}_1 \in \mathbb{R}^s, \|\boldsymbol{p}_1\| = 1$. Set $\beta_0 = 0$; |
| 3  **for** $i = 1, ..., l$ **do** | 3  **for** $i = 1, ..., l$ **do** |
| 4  $\quad \boldsymbol{q}_i \leftarrow \boldsymbol{Z}\boldsymbol{p}_i - \beta_{i-1}\boldsymbol{q}_{i-1}$; | 4  $\quad \boldsymbol{b}'_i \leftarrow (\sum_t \boldsymbol{p}_i[t] \cdot \boldsymbol{b}_t) - \beta_{i-1}\boldsymbol{b}'_{i-1}$; |
| 5  $\quad \alpha_i \leftarrow \|\boldsymbol{q}_i\|$; | 5  $\quad \boldsymbol{c}'_i \leftarrow (\sum_t \boldsymbol{p}_i[t] \cdot \boldsymbol{c}_t) - \beta_{i-1}\boldsymbol{c}'_{i-1}$; |
| 6  $\quad \boldsymbol{q}_i \leftarrow \boldsymbol{q}_i/\alpha_i$; | 6  $\quad \alpha_i \leftarrow \sqrt{\langle \boldsymbol{b}'_i, \boldsymbol{b}'_i \rangle - \langle \boldsymbol{c}'_i \boldsymbol{c}'_i \rangle}$; |
| 7  $\quad \boldsymbol{p}_{i+1} \leftarrow \boldsymbol{Z}^\top \boldsymbol{q}_i - \alpha_i \boldsymbol{p}_i$; | 7  $\quad \boldsymbol{b}'_i \leftarrow \boldsymbol{b}'_i/\alpha$; |
| 8  $\quad$ Orthogonalze $\boldsymbol{p}_{i+1}$ against $[\boldsymbol{p}_1, ..., \boldsymbol{p}_i]$; | 8  $\quad \boldsymbol{c}'_i \leftarrow \boldsymbol{c}'_i/\alpha$; |
| 9  $\quad \beta_i \leftarrow \|\boldsymbol{p}_{i+1}\|$; | 9  $\quad \boldsymbol{p}_{i+1} \leftarrow \boldsymbol{B}^\top \boldsymbol{b}'_i - \boldsymbol{C}^\top \boldsymbol{c}'_i - \alpha_i \boldsymbol{p}_i$; |
| 10  $\quad \boldsymbol{p}_{i+1} = \boldsymbol{p}_{i+1}/\beta_i$; | 10  $\quad$ Orthogonalze $\boldsymbol{p}_{i+1}$ against $[\boldsymbol{p}_1, ..., \boldsymbol{p}_i]$; |
| 11  **end** | 11  $\quad \beta_i \leftarrow \|\boldsymbol{p}_{i+1}\|$; |
| 12  $\boldsymbol{P}_{l+1} = [\boldsymbol{p}_1, ..., \boldsymbol{p}_{l+1}]$; | 12  $\quad \boldsymbol{p}_{i+1} = \boldsymbol{p}_{i+1}/\beta_i$; |
| 13  $\boldsymbol{H} = \mathrm{diag}\{\alpha_1, ..., \alpha_l\} + \mathrm{superdiag}\{\beta_1, ..., \beta_l\}$; | 13  **end** |
| 14  $\boldsymbol{P} \leftarrow \boldsymbol{P}_{l+1}\boldsymbol{H}^\top$; | 14  $\boldsymbol{P}_{l+1} = [\boldsymbol{p}_1, ..., \boldsymbol{p}_{l+1}]$; |
| 15  **return** $\boldsymbol{Q} = [\boldsymbol{q}_1, ..., \boldsymbol{q}_l], \boldsymbol{P}$ | 15  $\boldsymbol{H} = \mathrm{diag}\{\alpha_1, ..., \alpha_l\} + \mathrm{superdiag}\{\beta_1, ..., \beta_l\}$; |
|  | 16  $\boldsymbol{P} \leftarrow \boldsymbol{P}_{l+1}\boldsymbol{H}^\top$; |
|  | 17  **return** $\boldsymbol{Q} = (\boldsymbol{B}', \boldsymbol{C}'), \boldsymbol{P}$ |

**A step-by-step description.**  Specifically, the $\boldsymbol{Z}\boldsymbol{p}_i$ in Line 4 of Algorithm 5 can be viewed as a linear combination of SV-LCOV with Line 4 and Line 5 in Algorithm 6. The $l_2$ norm of $\boldsymbol{q}_i$ in Line 5 of Algorithm 5 is the length of a SV-LCOV and can be transformed into the inner product in Line 6 of Algorithm 6. Line 6 of Algorithm 5 is a scalar multiplication corresponding to Line 7 and Line 8 of Algorithm 6. And the $\boldsymbol{Z}^\top \boldsymbol{q}_i$ in Line 7 of Algorithm 5 can be recognized as the inner product between SV-LCOV demonstarted in Line 9 of Algorithm 6.

**Complexity analysis of Algorithm 6.**  Line 4, 5 run in $O((nnz(\boldsymbol{E}) + k)sl)$ time. Line 6 runs in $O(nnz(\boldsymbol{E} + k)l)$ time. Line 7, 8 run in $O((nnz(\boldsymbol{E}) + k)l)$ time. Line 9 runs in $O(nnz(\boldsymbol{E} + k)sl)$ time. Line 10 runs in $O(sl^2)$ time. Line 11, 12 run in $O(sl)$ time. Line 14, 15, 16 run in $O(sl^2)$ time.

The overall time complexity of Algorithm 6 is $O(nnz(\boldsymbol{E} + k)sl)$ time.

### D.2  APPROXIMATING THE AUGMENTED SPACE VIA RANDOM POWER ITERATION PROCESS

| **Algorithm 7:** RPI | **Algorithm 8:** RPI with SV-LCOV |
|---|---|
| **Input:** $\boldsymbol{E} \in \mathbb{R}^{m \times s}, \boldsymbol{U}_k \in \mathbb{R}^{m \times k}, l, t \in \mathbb{N}^+$ | **Input:** $\boldsymbol{E} \in \mathbb{R}^{m \times s}, \boldsymbol{U}_k \in \mathbb{R}^{m \times k}, l, t \in \mathbb{N}^+$ |
| **Output:** $\boldsymbol{Q} \in \mathbb{R}^{n \times l}, \boldsymbol{P} \in \mathbb{R}^{s \times l}$ | **Output:** $\boldsymbol{Q} \in \mathbb{R}^{n \times l}, \boldsymbol{P} \in \mathbb{R}^{s \times l}$ |
| 1  $\boldsymbol{Z} \leftarrow (\boldsymbol{I} - \boldsymbol{U}_k \boldsymbol{U}_k^\top)\mathbf{E}$; | 1  $\boldsymbol{B} \leftarrow \boldsymbol{E}, \quad \boldsymbol{C} \leftarrow \boldsymbol{U}_k^\top \boldsymbol{E}$; |
| 2  Choose $\boldsymbol{P} \in \mathbb{R}^{s \times l}$ with random unitary columns; | 2  Choose $\boldsymbol{P} \in \mathbb{R}^{s \times l}$ with random unitary columns; |
| 3  **for** $i = 1, ..., t$ **do** | 3  **for** $i = 1, ..., t$ **do** |
| 4  $\quad \boldsymbol{P}, \boldsymbol{R} \leftarrow QR(\boldsymbol{P})$; | 4  $\quad \boldsymbol{B}', \boldsymbol{C}' \leftarrow \boldsymbol{B}\boldsymbol{P}, \boldsymbol{C}\boldsymbol{P}$; |
| 5  $\quad \boldsymbol{Q} \leftarrow \boldsymbol{Z}\boldsymbol{P}$; | 5  $\quad \boldsymbol{B}', \boldsymbol{C}', \boldsymbol{R} \leftarrow QR$ with Algorithm 2; |
| 6  $\quad \boldsymbol{Q}, \boldsymbol{R} \leftarrow QR(\boldsymbol{Q})$; | 6  $\quad \boldsymbol{P}, \boldsymbol{R} \leftarrow QR(\boldsymbol{P})$; |
| 7  $\quad \boldsymbol{P} \leftarrow \boldsymbol{Z}^\top \boldsymbol{Q}$; | 7  $\quad \boldsymbol{P} \leftarrow \boldsymbol{B}^\top \boldsymbol{B}' - \boldsymbol{C}^\top \boldsymbol{C}'$; |
| 8  **end** | 8  **end** |
| 9  **return** $\boldsymbol{Q}, \boldsymbol{P}$ | 9  **return** $\boldsymbol{Q} = (\boldsymbol{B}', \boldsymbol{C}'), \boldsymbol{P}$ |

**Complexity analysis of Algorithm 8.**  Line 1 runs in $O(nnz(\boldsymbol{E})k^2)$ time. Line 4 runs in $O((nnz(\boldsymbol{E}) + k)slt)$ time. Line 5 runs in $O(nnz(\boldsymbol{E}) + k)l^2 t)$ time. Line 6 runs in $O(sl^2 t)$

time. Line 7 runs in $O((nnz(E) + k)slt)$ time. The overall time complexity of Algorithm 8 is $O((nnz(\boldsymbol{E}) + k)(sl + l^2)t + nnz(\boldsymbol{E})k^2)$.

### D.3 THE PROPOSED UPDATING TRUNCATED SVD WITH APPROXIMATE AUGMENTED MATRIX

---

**Algorithm 9:** Add columns with approximated augmented space

**Input:** $\boldsymbol{U}_k(\boldsymbol{U}', \boldsymbol{U}''), \boldsymbol{\Sigma}_k, \boldsymbol{V}_k(\boldsymbol{V}', \boldsymbol{V}''), \boldsymbol{E}$

1. Turn $(\boldsymbol{I} - \boldsymbol{U}_k\boldsymbol{U}_k^\top)\boldsymbol{E}$ into SV-LCOV and get $\boldsymbol{Q}(\boldsymbol{B}, \boldsymbol{C}), \boldsymbol{P}$ with Algorithm 6 or 8;

2. $\boldsymbol{F}_k, \boldsymbol{\Theta}_k, \boldsymbol{G}_k \leftarrow \text{SVD}_k(\begin{bmatrix} \boldsymbol{\Sigma}_k & \boldsymbol{U}_k^\top\boldsymbol{E} \\ & \boldsymbol{P}^\top \end{bmatrix})$;

3. $\boldsymbol{U}'' \leftarrow \boldsymbol{U}''(\boldsymbol{F}_k[:k] - \boldsymbol{C}\boldsymbol{F}_k[k:])$;

4. $\boldsymbol{U}' \leftarrow \boldsymbol{U}' + \boldsymbol{B}\boldsymbol{F}_k[k:]\boldsymbol{U}''^{-1}$;

5. $\boldsymbol{\Sigma}_k \leftarrow \boldsymbol{\Theta}_k$;

6. $\boldsymbol{V}'' \leftarrow \boldsymbol{V}''\boldsymbol{G}_k[:k]$;

7. Append new columns $\boldsymbol{G}[k:]\boldsymbol{V}''^{-1}$ to $\boldsymbol{V}'$;

---

**Algorithm 10:** Update weights with approximated augmented space

**Input:** $\boldsymbol{U}_k(\boldsymbol{U}', \boldsymbol{U}''), \boldsymbol{\Sigma}_k, \boldsymbol{V}_k(\boldsymbol{V}', \boldsymbol{V}''), \boldsymbol{D}, \boldsymbol{E}$

1. Turn $(\boldsymbol{I} - \boldsymbol{U}_k\boldsymbol{U}_k^\top)\boldsymbol{D}$ into SV-LCOV and get $\boldsymbol{Q}_D(\boldsymbol{B}_D, \boldsymbol{C}_D), \boldsymbol{P}_D$ with Algorithm 6 or 8;

2. Turn $(\boldsymbol{I} - \boldsymbol{V}_k\boldsymbol{V}_k^\top)\boldsymbol{E}$ into SV-LCOV and get $\boldsymbol{Q}_E(\boldsymbol{B}_E, \boldsymbol{C}_E), \boldsymbol{P}_E$ with Algorithm 6 or 8;

3. $\boldsymbol{F}_k, \boldsymbol{\Sigma}_k, \boldsymbol{G}_k \leftarrow$
$\text{SVD}_k(\begin{bmatrix} \boldsymbol{\Sigma}_k & \boldsymbol{0} \\ \boldsymbol{0} & \boldsymbol{0} \end{bmatrix} + \begin{bmatrix} \boldsymbol{U}_k^\top\boldsymbol{D} \\ \boldsymbol{P}_D^\top \end{bmatrix} \begin{bmatrix} \boldsymbol{V}_k^\top\boldsymbol{E} \\ \boldsymbol{P}_E^\top \end{bmatrix}^\top)$;

4. $\boldsymbol{U}'' \leftarrow \boldsymbol{U}''(\boldsymbol{F}_k[:k] - \boldsymbol{C}_D\boldsymbol{F}_k[k:])$;

5. $\boldsymbol{U}' \leftarrow \boldsymbol{U}' + \boldsymbol{B}_D\boldsymbol{F}_k[k:]\boldsymbol{U}''^{-1}$;

6. $\boldsymbol{\Sigma}_k \leftarrow \boldsymbol{\Theta}_k$;

7. $\boldsymbol{V}'' \leftarrow \boldsymbol{V}''(\boldsymbol{G}_k[:k] - \boldsymbol{C}_E\boldsymbol{G}_k[k:])$;

8. $\boldsymbol{V}' \leftarrow \boldsymbol{V}' + \boldsymbol{B}_E\boldsymbol{G}_k[k:]\boldsymbol{U}''^{-1}$;

---

# E EXPERIMENTS

## E.1 RUNTIME OF EACH STEP

We present the runtime analysis of each component in the experiments, specifically focusing on the verification of $\phi$ using the Slashdot datasets. Since all the baselines, as well as the proposed method, can be conceptualized as a three-step algorithm outlined in Section 2.1, we provide an illustration of the runtime for each step. Specifically, we break down the entire algorithm into three distinct steps: the stage before the execution of the compact SVD, the actual execution of the compact SVD, and the segment after the execution of the compact SVD. The experimental results are depicted in Figure 3.

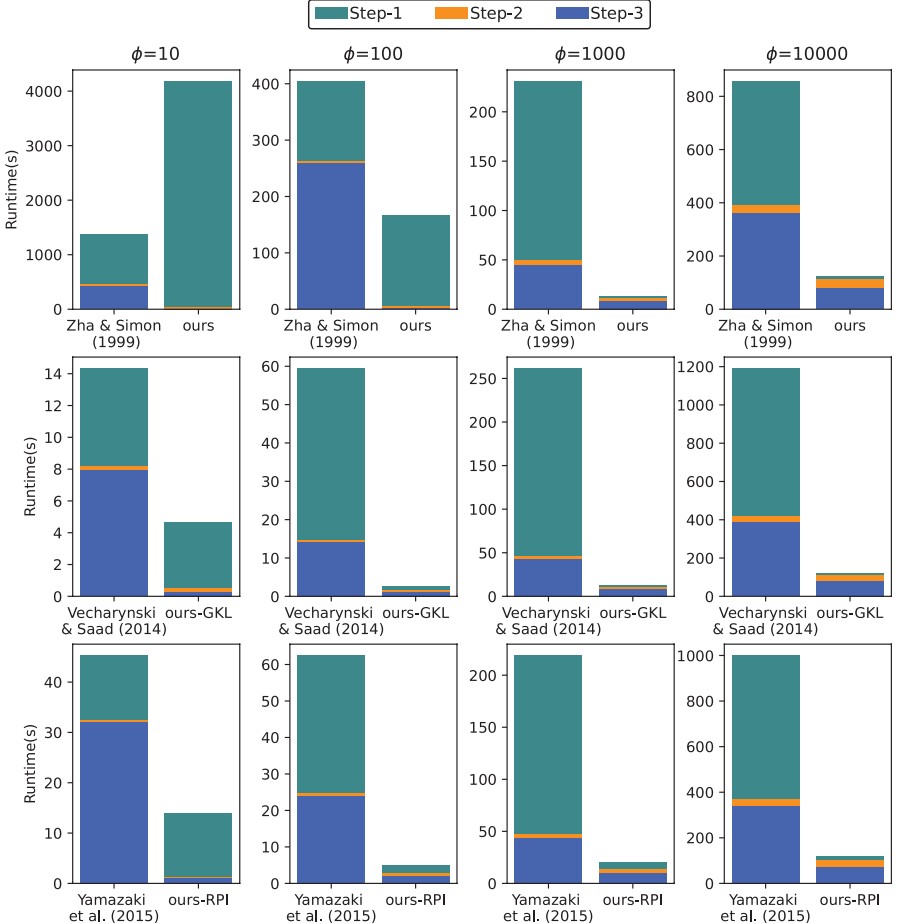

Figure 3: Runtime of each step on the Slashdot dataset.

Compared to the original methods, the proposed methods take approximately the same amount of time to execute the compact SVD. This similarity arises because both the proposed and original methods involve decomposing a matrix of the same shape. Furthermore, as the value of $\phi$ increases and $s$ decreases, the proportion of total time consumed by step-2 (compact SVD) increases. This trend suggests a more significant improvement in efficiency for step-1 and step-3 of the algorithm.

The proposed method mainly focuses on time complexity in step-1 and step-3, respectively benefiting from the structure of SV-LCOV described in Section 3.1 and the extended decomposition in Section 3.2. The optimization in the first step is more pronounced when $\phi$ is larger (i.e., when $s$ is smaller). This is due to the fact that in the SV-LCOV framework, the sparse vectors have fewer non-zero rows when $s$ is smaller. This reduction in non-zero rows is a consequence of fewer columns

in the matrix $\boldsymbol{E}$, resulting in more significant efficiency improvements through sparse addition and subtraction operations.

Furthermore, when $\phi$ is smaller and $s$ is larger, the utilization of the proposed variant method, which involves approximating the basis of the augmented space instead of obtaining the exact basis, tends to yield greater efficiency.

### E.2 Experiments on Synthetic Matrices with Different Sparsity

We conduct experiments using synthetic matrices with varying sparsity levels (i.e., varying the number of non-zero entries) to investigate the influence of sparsity on the efficiency of updating the SVD. Specifically, we first generate several random sparse matrices with a fixed size of $100{,}000 \times 100{,}000$. The number of non-zero elements in these matrices ranges from 1,000,000 to 1,000,000,000 (i.e., density from $0.01\%$ to $10\%$). We initialize a truncated SVD by utilizing a matrix comprised of the initial 50% of the columns. Subsequently, the remaining 50% of the columns are incrementally inserted into the matrix in $\phi$ batches, with each batch containing an equal number of columns. The number of columns added in each batch is denoted as $s$. The experimental results are shown in Figure 4.

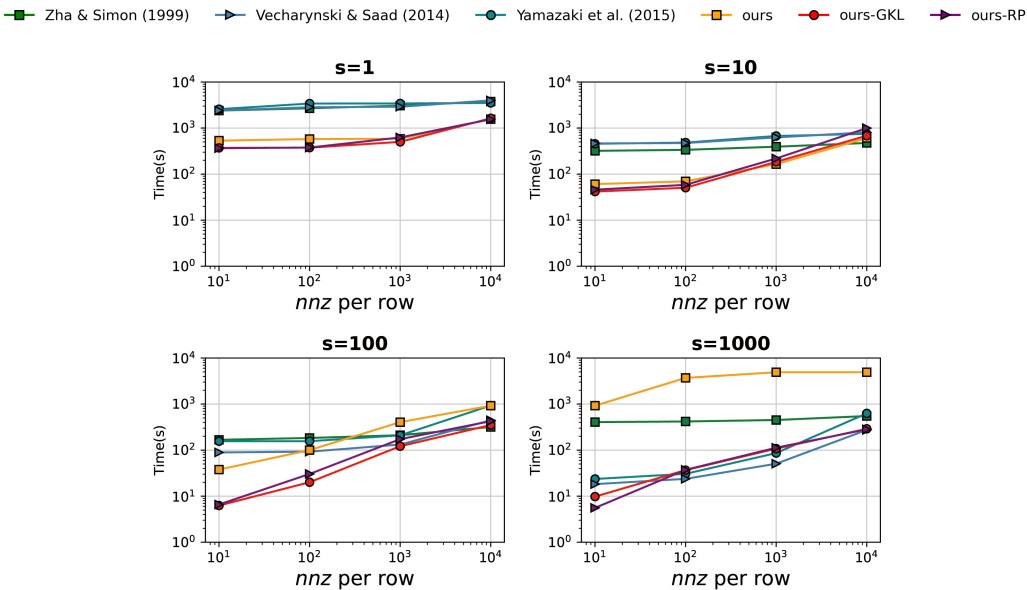

Figure 4: Computational efficiency on synthetic matrices

Experimental results under various $s$ values show that: 1) The proposed method exhibits greater efficiency improvement when the matrix is relatively sparse. 2) When the rank of the update matrix is higher, the variant using Lanczos vectors for space approximation achieves faster performance.

### E.3 Effectiveness

We report the Frobenius norm beteween $\boldsymbol{U}_k \Sigma_k \boldsymbol{V}_k^{\top}$ and original matrix in the Slashdot datasets and MovieLen25M datasets in Table 5. Our proposed approach maintains a comparable precision to baselines.

Table 5: Frobenius norm w.r.t $\phi$

| Method | Slashdot | | | MovieLen25M | | |
|---|---|---|---|---|---|---|
| | $\phi = 10^1$ | $\phi = 10^2$ | $\phi = 10^3$ | $\phi = 10^1$ | $\phi = 10^2$ | $\phi = 10^3$ |
| Zha & Simon (1999) | 784.48 | 792.16 | 792.11 | 4043.86 | 4044.23 | 4044.40 |
| Vecharynski & Saad (2014) | 802.61 | 796.26 | 792.01 | 4073.41 | 4111.66 | 4050.53 |
| Yamazaki et al. (2015) | 796.97 | 795.94 | 792.11 | 4098.87 | 4098.62 | 4047.87 |
| ours | 784.48 | 792.16 | 792.11 | 4043.86 | 4044.23 | 4044.40 |
| ours-GKL | 802.85 | 795.65 | 792.01 | 4076.61 | 4110.71 | 4050.36 |
| ours-RPI | 796.65 | 795.19 | 792.11 | 4099.11 | 4099.09 | 4047.20 |

## F  TIME COMPLEXITY ANALYSIS

A line-by-line analysis of time complexity for Algorithm 3, 4, 9, 10 is given below. Note that due to the extended decomposition, the time complexity of converting the augmented matrix into SV-LCOV is $O(nnz(\boldsymbol{E})k^2)$ instead of $O(nnz(\boldsymbol{E})k)$.

Table 6: Asymptotic complexity of Algorithm 3

| | Asymptotic complexity (big-$O$ notation is omitted) |
|---|---|
| Line 1 | $nnz(\boldsymbol{E})(k^2 + s^2)$ |
| Line 2 | $nnz(\boldsymbol{E})k^2 + (k + s)^3$ |
| Line 3 | $k^3 + k^2 s$ |
| Line 4 | $nnz(\boldsymbol{E})(ks + k^2) + k^3$ |
| Line 5 | $k$ |
| Line 6 | $k^3$ |
| Line 7 | $k^2 s + k^3$ |
| Overall | $nnz(\boldsymbol{E})(k + s)^2 + (k + s)^3$ |

Table 7: Asymptotic complexity of Algorithm 4

| | Asymptotic complexity (big-$O$ notation is omitted) |
|---|---|
| Line 1 | $nnz(\boldsymbol{D})(k^2 + s^2)$ |
| Line 2 | $nnz(\boldsymbol{E})(k^2 + s^2)$ |
| Line 3 | $nnz(\boldsymbol{D} + \boldsymbol{E})k^2 + (k + s)^3$ |
| Line 4 | $k^3 + k^2 s$ |
| Line 5 | $nnz(\boldsymbol{D})(ks + k^2) + k^3$ |
| Line 6 | $k$ |
| Line 7 | $k^3 + k^2 s$ |
| Line 8 | $nnz(\boldsymbol{E})(ks + k^2) + k^3$ |
| Overall | $nnz(\boldsymbol{D} + \boldsymbol{E})(k + s)^2 + (k + s)^3$ |

Table 8: Asymptotic complexity of Algorithm 9

|  | Asymptotic complexity (big-$O$ notation is omitted) |
|---|---|
| Line 1 (GKL) | $nnz(\boldsymbol{E})k^2 + nnz(\boldsymbol{E} + k)sl$ |
| Line 1 (RPI) | $(nnz(\boldsymbol{E}) + k)(sl + l^2)t + nnz(\boldsymbol{E})k^2$ |
| Line 2 | $(k + s)(k + l)^2$ |
| Line 3 | $k^3 + k^2l$ |
| Line 4 | $nnz(\boldsymbol{E})(kl + k^2) + k^3$ |
| Line 5 | $k$ |
| Line 6 | $k^3$ |
| Line 7 | $k^2l + k^3$ |
| Overall (GKL) | $nnz(\boldsymbol{E})(k^2 + sl + kl) + (k + s)(k + l)^2$ |
| Overall (RPI) | $nnz(\boldsymbol{E})(sl + l^2)t + nnz(\boldsymbol{E})k^2 + (k + s)(k + l)^2$ |

Table 9: Asymptotic complexity of Algorithm 10

|  | Asymptotic complexity (big-$O$ notation is omitted) |
|---|---|
| Line 1 (GKL) | $nnz(\boldsymbol{D})k^2 + nnz(\boldsymbol{D} + k)sl$ |
| Line 1 (RPI) | $(nnz(\boldsymbol{D}) + k)(sl + l^2)t + nnz(\boldsymbol{D})k^2$ |
| Line 2 (GKL) | $nnz(\boldsymbol{E})k^2 + nnz(\boldsymbol{E} + k)sl$ |
| Line 2 (RPI) | $(nnz(\boldsymbol{E}) + k)(sl + l^2)t + nnz(\boldsymbol{E})k^2$ |
| Line 3 | $nnz(\boldsymbol{D} + \boldsymbol{E})k^2 + (k + l)^3$ |
| Line 4 | $k^3 + k^2l$ |
| Line 5 | $nnz(\boldsymbol{D})(kl + k^2) + k^3$ |
| Line 6 | $k$ |
| Line 7 | $k^3 + k^2l$ |
| Line 8 | $nnz(\boldsymbol{E})(kl + k^2) + k^3$ |
| Overall (GKL) | $nnz(\boldsymbol{D} + \boldsymbol{E})(k^2 + sl + kl) + (k + s)(k + l)^2$ |
| Overall (RPI) | $nnz(\boldsymbol{D} + \boldsymbol{E})(sl + l^2)t + nnz(\boldsymbol{D} + \boldsymbol{E})k^2 + (k + s)(k + l)^2$ |

## G  ALGORITHM OF ADDING ROWS

---
**Algorithm 11:** Add rows
---
**Input:** $\boldsymbol{U}_k(\boldsymbol{U}', \boldsymbol{U}''), \boldsymbol{\Sigma}_k, \boldsymbol{V}_k(\boldsymbol{V}', \boldsymbol{V}''), \boldsymbol{E}_r$

1 Turn $(\boldsymbol{I} - \boldsymbol{V}_k\boldsymbol{V}_k^\top)\boldsymbol{E}_r^\top$ into SV-LCOV and get $\boldsymbol{Q}(\boldsymbol{B}, \boldsymbol{C}), \boldsymbol{R}$ with Algorithm 2;

2 $\boldsymbol{F}_k, \boldsymbol{\Theta}_k, \boldsymbol{G}_k \leftarrow \text{SVD}_k(\begin{bmatrix} \boldsymbol{\Sigma}_k \\ \boldsymbol{E}_r^\top\boldsymbol{V}_k & \boldsymbol{R}^\top \end{bmatrix})$;

3 $\boldsymbol{U}'' \leftarrow \boldsymbol{U}''\boldsymbol{F}_k[:k]$;

4 Append new rows $\boldsymbol{F}[k:]\boldsymbol{U}''^{-1}$ to $\boldsymbol{U}'$;

5 $\boldsymbol{\Sigma}_k \leftarrow \boldsymbol{\Theta}_k$;

6 $\boldsymbol{V}'' \leftarrow \boldsymbol{V}''(\boldsymbol{G}_k[:k] - \boldsymbol{C}\boldsymbol{G}_k[k:])$;

7 $\boldsymbol{V}' \leftarrow \boldsymbol{V}' + \boldsymbol{B}\boldsymbol{G}_k[k:]\boldsymbol{V}''^{-1}$;

---

