# OpenReview forum: "Fast Updating Truncated SVD for Representation Learning with Sparse Matrices"
_ICLR.cc/2024/Conference — ICLR 2024 poster_

### Official Review · Reviewer_uuTf · 2023-10-30

**Soundness:** 3 good
**Presentation:** 2 fair
**Contribution:** 3 good
**Rating:** 8
**Confidence:** 2

**Summary:**

This paper presents an improved method for the computation of truncated SVD by exploiting the sparsity of the involved matrix. Here this work focus on evolving matrices and updates of a current SVD. The proposed framework shows lesser complexity than previous methods with good accuracy. The problematic is fairly general and such framework can be of used in many situations.

**Strengths:**

The proposed framework enjoys better complexity which keeping a good precision. This is especially interesting while dealing with a stream of data. The experiments show the efficiency of the framework for different tasks.

**Weaknesses:**

The paper is not always easy to read. For example, the contribution are hard to clearly assess at first even if they are detailled at the beginning of the paper.

I find on minor type in page 3: "Orthogonalzation" instead of "Orthogonalization".

**Questions:**

The precision of the computation is mentioned on the beginning of the paper but no theorem, proposition or lemma support the claim. Is it possible to prove some error bound?

The "Add rows" algorithm is missing, I assume that it is very close to the "Add columns" algorithm. Please clarify.

---

> ### Author Response · Authors · 2023-11-21
> **Response to Reviewer uuTf**
>
> We appreciate the comments provided by the reviewer. Thank you for taking the time to review our work.
>
> We have revised the description in the theorem to provide a clearer explanation of our 'approximate' approach. Our proposed method maintains the same error bound as the work of Zha & Simon. For a detailed error analysis, we refer readers to references [1] and [2].
>
> The "Add rows" algorithm is symmetric to the "Add columns" algorithm. In the revised version, we have included the pseudocodes for both algorithms in the Appendix to provide a comprehensive understanding of their implementation.
>
> [1] Zha, Hongyuan, and Horst D. Simon. "On updating problems in latent semantic indexing." *SIAM Journal on Scientific Computing* 21.2 (1999): 782-791.
>
> [2] Zha, Hongyuan, and Zhenyue Zhang. "Matrices with low-rank-plus-shift structure: partial SVD and latent semantic indexing." *SIAM Journal on Matrix Analysis and Applications* 21.2 (2000): 522-536.

---

> ### Comment · Reviewer_uuTf · 2023-11-22
> **Response to the authors**
>
> Thank you for the clarification. I read the different reviews and I now better understand the contribution of this paper. I will my rating to 8.

---

> ### Author Response · Authors · 2023-11-23
>
> Thank you. We are sincerely grateful to your appreciation of our work.

---

### Official Review · Reviewer_YLvs · 2023-10-31

**Soundness:** 3 good
**Presentation:** 3 good
**Contribution:** 3 good
**Rating:** 6
**Confidence:** 5

**Summary:**

This paper presents a new algorithm to update the truncated SVD of a dynamical matrix (new rows, columns, and/or entries).
The algorithm is similar to that of previous algorithms published in the literature. The main innovation seems to be a new data
structure that allows a more efficient handling and processing of intermediate matrices (by essentially not forming them explicitly).

**Strengths:**

-) All concepts and ideas are clear and the paper is not verbose. Codes and appendices are also provided.
-) Results suggest good performance although I do have concerns (see below).
-) The topic is important and interesting.

**Weaknesses:**

-) It is not clear whether the superiority of the new algorithm, as evidenced by the numerical results, is natural or the result of inefficient comparisons. See my comments below.
-) While the algorithmic contribution is non-trivial, the new algorithms are not really new since they follow what other published algorithms already do, albeit in a more efficient fashion.

**Questions:**

-) It seems that GKL helps the algorithm proposed by the authors but does not help the method of Vecharynski (in terms of performance). What is the exact process followed when applying GKL to the latter? How is the method of Vecharynski slower for smaller values of k (k=16 versus k=64)?

-) I am a bit puzzled since the new method replicates existing methods but is an order of magnitude faster. Is this really because of the data structure?

-) Upon inspecting the code, I see that the authors form the matrix ‘X’ explicitly when running the methods of Vecharynski and Yamazaki. This implementation is quite inefficient and now I understand better why the new method is that much faster. While for the method of Zha and Simon I can understand forming ‘X’ explicitly (even then it can be implemented more appropriately though) there is absolutely no reason to form ‘X’ for GKL (or randomized SVD) since the main advantage of GKL is that it can be applied to implicitly-defined matrices. Forming ‘X’ creates a huge dense matrix that is a) expensive to apply, b) expensive to store in system memory. For large problems, one might have to store ‘X’ on secondary memory, leading to quite high wall-clock time execution such as the ones I see in this submission. All numerical experiments concerning timings must be performed from scratch using commonly accepted principles of implementing numerical algorithms.

Also, the code would be benefit tremendously by adding comments.

-) The method of Zha and Simon should be more accurate than the algorithms of Vecharynski and Yamazaki, yet it seems that all three methods basically give the same accuracy. One reason for that is when ‘l’ is too large. What are the results when ‘l’ is smaller?

-) At the beginning of page 7, the paragraph “As a result...” is a bit strange. Is any text missing?

-) “trunacated” -> “truncated”

-) Finally, a general comment: "In Table 1, the method of Kalantzis et al. depends on nnz(A) because the matrix A is fetched at the end of each update to compute the right singular vectors in a least-squares sense. The method can be asymptotically more expensive but it is also more accurate."

Update after response:
I have altered my score.

---

> ### Author Response · Authors · 2023-11-21
> **Response to Reviewer YLvs**
>
> We would like to extend our sincere gratitude to you for their invaluable comments and feedback.
>
> We have retested the runtime of Vecharynski's method under different values of $k$, and have rectified the results. It appears that we mistakenly swapped the two times in our previous reporting. We sincerely apologize for this oversight.
>
> **[Implement of baseline]**
>
> We have made revisions to the implementation of this section and reevaluated the runtime accordingly. Instead of directly constructing $X$ (i.e., $(I-U_kU_k^\top)E$), we access it through matrix-vector multiplication. This improvement significantly enhances efficiency, particularly in scenarios where $\phi$ is smaller (i.e., with larger update granularity). However, when $\phi$ is larger, the efficiency decreases. For both Vecharynski and Yamazaki methods, we have considered the lesser of the two times as the final result and have accordingly updated all experimental findings. The revised version of the paper now includes the related description as suggested. We will soon update the modified code in the anonymous repository.
>
> Thank you for reviewing our code. We will follow your advice to add comments to enhance the readability of the code.
>
> **[Accuracy of smaller $\phi$]**
>
> We have included the Frobenius norm values of the original and reconstructed matrices for various $\phi$ values in Appendix E.3. Experimental results demonstrate that our proposed approach significantly enhances efficiency while maintaining comparable precision to previous methods.
>
> Comparatively, for smaller values of $\phi$ , the algorithms of Vecharynski and Yamazaki demonstrate a higher degree of accuracy loss compared to the algorithm of Zha & Simon. We suspect that this discrepancy could be attributed to larger values of $s$  when $\phi$ is smaller in our experiment. These two methods approximate a higher-dimensional space with a significantly lower-dimensional basis. From the perspective discussed in [1],  it is possible that these methods might have difficulty accurately capturing the $range(U_k)$, leading to a loss of accuracy.
>
> **[Visualize runtime of each step]**
>
> The baselines, the proposed method, and its variants in the experiment can all be conceptualized as onsisting of three steps outlined in Section 2.1 (i.e., the pre-processing stage before conducting the compact matrix SVD, the compact matrix SVD process, the post-processing stage). We have generated visualizations to showcase the runtime of each step for these methods under different 𝜙 values.
>
> The experimental results presented in Appendix E.1 demonstrate that the proposed method exhibits efficiency improvements in the first and third steps. These improvements can be attributed to the novel structure employed for the basis of the augmented matrix and the additional decomposition step, respectively.
>
> **[Writing]**
>
> In the revised version of the paper, we have addressed and corrected several typos and grammatical errors. Additionally, we have made clarifications in areas that may have caused ambiguity.
>
> [1] Kalantzis, Vasileios, et al. "Projection techniques to update the truncated SVD of evolving matrices with applications." *International Conference on Machine Learning*. PMLR, 2021.

---

### Official Review · Reviewer_jDYv · 2023-10-31

**Soundness:** 3 good
**Presentation:** 1 poor
**Contribution:** 3 good
**Rating:** 6
**Confidence:** 4

**Summary:**

This paper presents a method for updating the truncated singular value decompositions for sparse updates. The "trick" of this paper is a "sparse vector minus a linear combination of orthonormal vectors" representations, which is used to represent the orthogonalization of the newly inducted rows or columns against an existing basis of singular vectors. Numerical experiments show that the proposed methods leads to order-of-magnitude speedups for stylized machine learning tasks.

**Strengths:**

The paper proposes a simple and natural idea. The truncated SVD is a useful computational tool in data analysis, and the proposed approach is a natural and effective way of updating the approximation under sparse updates of the matrix. The numerical experiments are reasonably convincing, and they demonstrate a large speedup over existing approaches.

**Weaknesses:**

This paper has two weaknesses I would like to discuss: issues of writing and grammar and potential numerical instabilities. I believe that the first of these can be addressed by revisions, and the second of these is an inherent (potential) limitation of the method that deserves a brief discussion by the authors in a revision.

### Writing

This paper has several issues with writing and presentation.

Throughout, there are issues of grammar and English usage. The issues are numerous, but here are a few examples that stand out:

- The title and the first sentence of the abstract do not parse as grammatically correct.
- Several key phrases are ungrammatical: in many places, "truncated SVD" should be replaced by "_the_ truncated SVD" (e.g., "updating truncated SVD" should be "updating _the_ truncated SVD"). The "augment matrix" should be "augmented matrix". "Augment procedure" should be "augmentation procedure". Etc.
- There are other grammatical issues. For instance, consider the following sentence: "A series of methods that can be recognized as an instance of Rayleigh-Ritz projection has become a mainstream method owing to its high precision." The verb "has" should be changed to "have" as the subject of the sentence "series of methods" is plural. Similarly, "its" should be "their". Even with grammatical fixes, the sentences are difficult to read. A better version of the sentence would be as follows: "In the past twenty-five years, Rayleigh-Ritz projection methods have become the standard methods for updating the truncated SVD, owing to their high accuracy."

The grammatical issues are serious enough to make the paper more difficult to read and understand. I recommend the authors do thorough revisions of their paper to improve the grammar and English usage.

There are also structural and clarity issues with the writing. Again, there are many issues, of which we highlight a few:

- The phrase, "augment matrix" (which should be changed for grammar to "augmented matrix") appears in section 1 before that term is defined or even the problem stated.
- The authors state that they provide "RPI and GKL" variants of their algorithm in section 3.3, but these acronyms were defined only briefly earlier in section 3.1—a section I skimmed on my first reading. A backward reference GKL and RPI, e.g., "(see section 3.1)" would help readers.
- The precise meaning of isometric in lemma 3 is unclear.
- As far as I can tell, the average precision metric is never defined.

The paper could benefit from reorganization to improve the narrative and sequencing of ideas.

Lastly, there's an issue of framing of the truncated SVD updating problem. In section 2, the present the problem they're solving as "approximating the truncated SVD of $\overline{A} = [A,E]$". However, the problem that is really being solved is to compute (exactly) the truncated SVD of $[\hat{A},E]$, where $\hat{A} = U\Sigma V^\top \approx A$ is the truncated SVD of $A$. The original Zha–Simon procedure is very clear about this distinction, writing

> Notice that in all the above three cases instead of the original term-document matrix $A$, we have used $A_k$, the best rank-$k$ approximation of $A$ as the starting point of the updating process. Therefore, we may not obtain the best rank-$k$ approximation of the true new term-document matrix."

I think the authors would benefit from this level of clarity about exactly what problem they are solving.

Overall, this paper would be greatly improved by rewriting to improve grammar and clarity.

### Numerical Stability?

There are several aspects of the current proposal that are concerning from the standpoint of numerical stability. The proposed orthogonalization procedure uses the Gram–Schmidt process, which is well-known to suffer from potentially significant loss of orthogonality. Additionally, the updating rules decompose a matrix $U_k$ with orthonormal columns are the product of two matrices, both of which could become significantly ill-conditioned and resulting in stability degradations.

Assuming these stability issues are real, they are intrinsic limitations to the method; it's very unclear how these issues could be addressed while maintaining the method's competitive runtime. The numerical results suggest that the method is stable enough to remain useful for some data analysis tasks. I would like to see the authors comment on numerical stability in any revised version, just to mention this as a potential issue with the method.

**Questions:**

### Typos and Minor Issues

There are small typos throughout, and I encourage the authors to reader their paper carefully to catch all of them. Here were a few that I happened to document:

- On the top of pg. 3, $U_kU_k$ and $V_kV_k$ should have transposes on the second factors.
- "Orthogonalization" is misspelled in a section header on page 3.
- There's a typo in (3): the middle line should have $U_kC$ not $U_k^\top C$.
- There are inconsistencies in typesetting. E.g., some A's are set in upright bold font and others are in slanted bold font.

---

> ### Author Response · Authors · 2023-11-21
> **Response to Reviewer jDYv**
>
> We appreciate the reviewer's valuable comments. Thank you for the suggestion, and we have revised the writing and addressed the grammar issues in the paper revision.
>
> We have partially restructured the paper, and in the revised version, we supplemented missing definitions, including those for augmented matrix, isometric, and average precision, which are highlighted in blue.
>
> **[Numerical Issue of QR factorization]**
>
> We have replaced the Gram-Schmidt process in Section 3.1 with the modified Gram-Schmidt process to enhance the numerical stability.
>
> The GS process demonstrates relative stability when the value of $s$ is small. In practical applications, Lanczos vectors are commonly employed to approximate the basis for efficiency considerations, especially when the value of *s* is large. Therefore, we recommend utilizing the proposed variant method when dealing with large values of $s$.
>
> **[Numerical Issue of the extended decomposition]**
>
> The condition number of the k-by-k matrix in the extended decomposition is close to $1$ in our experiments. In the following results, we present the maximum condition number observed throughout the entire update process for different numbers of update batches (i.e. $\phi$) on the Slashdot dataset. (Please note that in this experiment, the value of $s$ is inversely proportional to $\phi$.)
>
> | $\phi$   | $10^1$ | $10^2$ | $10^3$ | $10^4$ |
> | -------- | ------ | ------ | ------ | ------ |
> | ours     | 1.745  | 1.671  | 1.630  | 1.620  |
> | ours-GKL | 1.518  | 1.250  | 1.226  | 1.167  |
> | ours-RPI | 1.137  | 1.138  | 1.143  | 1.149  |
>
> The experimental results show that the condition numbers of the matrices remain relatively stable even after numerous updates, exhibiting no exponential or significant growth.
>
> In the body of the Section 3.2, we have provided further elaboration by discussing potential errors that could be caused by this aspect. In practice, the problem of matrices with large condition numbers can be solved by multiplying $U^\prime$ and $U^{\prime \prime}$ and resetting the resulting $k$-by-$k$ matrix to the identity. Considering the time efficiency improvement provided by the proposed method, we believe this error is acceptable.

---

> > ### Comment · Reviewer_jDYv · 2023-11-22
> >
> > I continue to feel that the writing issues are quite significant and far from being fully fixed in the current draft. I strongly advise the authors continue to make revisions, bringing it additional help for editing for grammar and phrasing if necessary.
> >
> > I would also like to see more discussion of numerical issues in either the text or supplement and how MGS partially, though not fully, addresses these concerns. (Mentioning An3v's Cholesky QR suggestion in this context seems appropriate.) Also, the scope could be more clearly stated in terms of just how sparse the updates need to be to contain a computational speedup.
> >
> > Overall, I still think this is a nice cute idea. It wasn't obvious to me, and it may prove useful in some applications. I maintain my score (with the understanding that the authors will continue to make significant efforts to improve the writing, as they have agreed to do in their global response above).

---

> > > ### Author Response · Authors · 2023-11-23
> > >
> > > Thank you for providing feedback on our paper. We appreciate your concerns regarding writing issues in the current draft and are committed to enhancing clarity and quality through additional revisions.

---

### Official Review · Reviewer_An3v · 2023-11-01

**Soundness:** 2 fair
**Presentation:** 1 poor
**Contribution:** 2 fair
**Rating:** 5
**Confidence:** 4

**Summary:**

This manuscript proposes a scheme for computing the QR factorization of a matrix of the form $(I-UU^T)B$ for a sparse $B$ that is more efficient than the naive approach. The scheme is then used within existing schemes for updating truncated SVDs to accelerate them.

**Strengths:**

The manuscript identifies a place for improvement (in specific settings, see the weaknesses section) within existing algorithms for updating truncated SVDs and implements an updated orthogonalization routine that accelerates them. Numerical experiments show the potential efficacy of the scheme.

**Weaknesses:**

This manuscripts largest weakness (and a significant one) is its lack of clarity in many respects. This spans the interplay with existing algorithms, the contribution, in what setting the proposed method is faster, and more. While the former two points could be addressed (though the manuscript does need significant reworking), the clarity around the appropriate settings for the algorithms, and therefore the contribution, is much less clear. This is most easily shown by illustration:

To choose one of the settings, let's consider updating (i.e., adding) columns to $A.$ In this case a QR factorization of $(I-U_kU_k^T)E$ is desired. The manuscript assumes $E$ is sparse, however the improvement is actually in a narrower regime. If $E$ has a constant number of non-zeros per row then nnz(E) is $\mathcal{O}(n)$ and the algorithms discussed in the manuscript are no more efficient than just forming $E - U(U^TE)$ and computing the dense QR factorization $\mathcal{O}(ns^2)$. So, the only setting of interest is if $E$ has a constant number of non-zeros per column, i.e., $\mathcal{O}(s)$ non-zeros. In this setting relatively few rows of $A$ receive updates. I don't know which setting is more/less common, but the manuscirpt lacks a clear articulation of such tradeoffs (including in the numerical experiments). This weakens the manuscript and makes it hard to understand what settings it applies to.

Similarly, it is not clear that most of section 3.1 is necessary. Bluntly it seems like an overly complicated build up of a simple choice. Considering the above setting, if $E$ has a constant number of non-zeros per column then why not just compute the Cholesky factorization $E^TE-(E^TU_k)(U_k^TE) = RR^T$ (which is \mathcal{O}(s^3) under this sparsity assumption) and then store the QR factorization as $(I-U_kU_k^T)(ER^{-1})$ (again, computing $ER^{-1}$ via triangular solves is only $\mathcal{O}(s^3)$ given the sparsity). Note that these all could be written with nnz(E), but again this is all only needed if nnz(E) is substantially less than $n.$ The method already implicitly assumes $U_k$ is stored as part of the process. In some sense this is what the manuscript proposes, but with what feels like an unnecessary amount of machinery that obscures the message. (Also, use of Cholesky would likely be faster since standard libraries could be used throughout). Moreover, given this simple interpretation it is not clear that Theorem 1 is particularly novel or interesting.

Related to the above points, the numerical experiments would be stronger if they were more clearly designed to show the tradeoffs between methods (e.g., using some synthetic examples where the sparsity can be controlled) rather than just "plausible" situations. The former actually seems more important since the accuracy should be the same. The key is to show when to use which method base on, e.g., the sparsity of $E.$ The numerical experiments are not illuminating in this regard.

While there is clearly effort in blending these ideas into prior work and building the numerical experiments (and therefore some contribution), ultimately the manuscript does not do a good job clearly articulating what this is. This is compounded by a lack of clarity in many places.

Assorted minor comments:

- Maybe it would be better to use different notation for the row, column, and low-rank updates to $A$ (i.e., not always using $E$). This can actually get confusing when thinking about dimensions of the appropriate projections.

- Equation (3) has a typo, it should be $U_kC.$

The paper has numerous grammatical mistakes and typos, for example in the title, abstract, and first paragraph of the intro:

- in the title what is "in sparse matrix"? The statement does not make sense, is it supposed to be "with sparse matrices"?

- abstract: "updating truncated singular value decomposition" -> "updating a truncated singular value decomposition"

- abstract: "Numerical experimental results on updating truncated SVD for evolving sparse matrices" -> Numerical experiments updating a truncated SVD for evolving sparse matrices

- abstract: "maintaining precision comparing" -> "maintaining precision comparable"

- intro P1: "Truncated Singular Value Decomposition (truncated SVD) is widely used in various machine..." -> "Truncated Singular Value Decompositions (SVDs) are widely used in various machine..."

- intro P1: "learning with truncated SVD benefits..." -> "learning with a truncated SVD benefits"

- intro P1: "interpretability provided by optimal Frobenius-norm" is not a cogent statement about the SVD; maybe what is meant is "interpretability because of its optimal approximation properties"? or similar?

- intro P1: "data under randomized" -> "data using randomized"

Accordingly, the manuscript would greatly benefit from a careful editing pass to address these issues (as they continue throughout the manuscript).

**Update after author response**

I would like to thank the authors for their thoughtful replies to my concerns and those of other reviewers.

While some of the responses do help highlight the scope of the contribution (e.g., in terms of how to think of nnz(E)), this is not really reflected in the manuscript. Saying “input sparsity” time is fine in places, but it is more useful to follow up and contextualize that statement in terms of the sparsity of the update. This does not really seem present in the updated manuscript.

Nevertheless, my overall opinion of the manuscript remains essentially unchanged (I have slightly updated my score, though the unallowable 4 would more accurately reflect my current opinion). There is a (somewhat narrow, but perhaps sufficient) contribution and it might be a good fit for certain applications. However, the presentation of the manuscript is still lacking and that significantly blunts any contribution. Numerous grammatical errors remain, Theorem 1 is still not sensibly stated (i.e., Definition 2 does not actually fix my concern and “approximate” is still ambiguously defined) to be a sound theoretical result, Section 3.1 could be more simply presented, and there remain aspects of the manuscript that are unclear (some of which are new additions). E.g., the paragraph above Alg 1 and 2 is not incorrect (i.e., it refers to the old version, Appendix E.2 doesn’t even say what the experiment is (updating rows, updating columns, updating weights, …), In Section 3.3 there is an incorrect algorithm reference, which makes Appendix F unclear (is the complexity of add columns or add rows analyzed), and more.

Lastly, on a technical note, while the use of MGS as suggested addresses one type of potential numerical issue (i.e., if $B$ lies close to the column space of $U_k$), it does not address potential issues encountered by the persistent decoupled storage of vectors in the sparse + low-dimensional subspace. This may or may not be problematic. Hence, while Cholesky QR is also often ill-advised, it is not clear it is worse in this setting (and given some of the condition numbers reported later likely fine for the simple tasks).

**Questions:**

In section 3.2, why is $BF_k[k:]$ sparse? in general the singular vectors in $F$ will not be sparse so the product likely is not as well. I guess this sort of depends on if the sparsity of $B$ is assumed to be $\mathcal{O}(n)$ (i.e., a few non-zeros per row) or $\mathcal{O}(s)$ (i.e., a few non-zeros per column). In the former case the product is not sparse and in the latter it is. This should be made more clear.

Truncated SVD of what in equation (4)? This is not clear from the text as written. Also, how is the decomposition into, e.g., $U'$ and $U''$ done? That is also not made clear. Even if there are details in the reference it is not clear how that is used here. Maybe cite a specific algorithm or technique from that work.

- What is meant buy "approximate" in Theorem 1? This seems too ambiguous a phrase for a theorem statement and it does not seem to be well defined anywhere.

- Why Gram-Schmidt and not modified Gram-Schmidt? and are there any potential numerical issues, e.g., if columns of $B$ are nearly perpendicular to the span of $U_k$?

- In the comparisons with prior work how is the QR factorization of $(I-U_kU_k^T)E$ (for example) computed? is a built in routine used or GS? There are benefits to being able to use standard libraries that could make understanding the tradeoffs more complicated/nuanced.

---

> ### Author Response · Authors · 2023-11-21
> **Response to Reviewer An3v**
>
> Thank you for dedicating your time to reviewing our paper and providing feedback on the grammatical issues. We apologize for any confusion caused during your review. We have taken your comments seriously and made significant revisions to address issues highlighted in your review. We meticulously reviewed the entire manuscript, correcting grammatical errors and ensuring sentence clarity and accuracy.
>
> **[In what setting the proposed method is faster]**
>
> The insight of this paper is the elimination of the $O(n)$ and $O(m)$ terms from the time complexity (where n and m are the length and width of the matrix, respectively). We adopt this approach because, when updating large-scale matrices, the $O(n)$ or $O(m)$ term typically dominates the overall time complexity compared to other factors such as $k$ (the dimension of the decomposition), $s$ (the rank of the update), and $nnz$ (the number of non-zero rows or columns in the update matrix). Therefore, the proposed method exhibits significant improvement when the values of $s$ or $nnz$ are small. However, when $s$ or $nnz$ becomes comparable to $O(n)$ or $O(m)$ (i.e., the terms we aim to eliminate), the optimization effect diminishes.
>
> We conduct numerical experiments with synthetic matrices in **Appendix E.2**. Specifically, we generate several random sparse matrices with a fixed size of 100,000 * 100,000. The number of non-zero elements in these matrices ranges from 1,000,000 to 1,000,000,000 (i.e., density from 0.01% to 10%). Experimental results with various $s$ values show that: 1) The proposed method exhibits more substantial efficiency improvements when the matrix is relatively sparse. 2) When the rank of the update matrix is higher, the variant using Lanczos vectors for space approximation achieves faster performance.
>
> **[About Section 3.1]**
>
> In Section 3.1, we present the fundamental operations in an inner product space, which enable the proposed structure to enhance the efficiency of various previous works on SVD update within a unified framework. We dedicate the entire Section 3.1 to present these operations, highlighting that our approach not only enables efficient QR decomposition but also serves as a generalized framework for approximating space bases, such as the GKL and RPI processes. We believe it would be interesting to provide more fundamental definitions and examine how a series of processes within the space, such as GS, Modified GS, and Lanczos vectors, are optimized from a unified perspective. Due to the page limit of ICLR, we provide detailed explanations of these methods in the appendix.
>
> A faster QR process is not the only contribution of this paper. In fact, apart from enhancing efficiency, another important motivation for proposing the structure is to accommodate the extended decomposition presented in Section 3.2. By merging the low-dimensional components of the proposed structure through Equation (3), we achieve a reduction in the overall asymptotic time complexity.
>
> **[Sparsity of $BF_k[k:]$]**
>
> The number of non-zero entries of the $n$-by-$k$ matrix $BF_k[k:]$ is less than $nnz(B)\cdot k$.
>
> Here, 'sparse' is used to indicate that this addition should be a *sparse addition* to ensure the asymptotic time complexity, i.e., the time complexity of this addition step is $O(nnz(B) \cdot k)$, not $O(nk)$. For relatively dense matrices, the utilization of *sparse addition* in this context can still maintain the asymptotic time complexity.
>
> **[Extended decomposition and equation (4)]**
>
> We have included a description of of this part in the revised version.
>
> **["Approximate" in Theorem 1]**
>
> In the revised version, we have provided a clarification for this aspect. When calculating the new truncated SVD, we utilize a low-rank approximation of the original matrix as the initial reference, rather than using the original matrix itself.
>
> **[Numerical Issue of Gram-Schmidt]**
>
> Thank you for your suggestion. In the revised version, we have replaced Gram-Schmidt with the modified Gram-Schmidt. When a vector is parallel or nearly parallel to the preceding span, the result of the orthogonalized vector can be set to the zero vector. From the perspective of the Rayleigh-Ritz process, this situation can be regarded as the current space being capable of approximating the singular space of the new matrix well without requiring the addition of new dimensions. We have included this observation as a remark in the revised version.
>
> **[QR factorization in the previous work]**
>
> In the experiments, the baseline method employs the standard library for the QR process.

---

> ### Comment · Reviewer_An3v · 2023-11-22
> **Review updated**
>
> I am not sure if a notification gets sent out when an official review is updated (I did not get one), so this comment is to note that I have updated my official review after reading the author response.

---

> > ### Author Response · Authors · 2023-11-23
> >
> > Thank you for your comments. We sincerely appreciate your valuable feedback. We have carefully noted the issues you raised and are diligently working on addressing them in the latest version. We will make the necessary revisions based on your suggestions to improve the paper.
> >
> > We have considered and attempted to employ the Cholesky QR approach. In situations where the update matrix is rank deficient, we substituted Cholesky QR with an eigen decomposition. It is common for the update matrix to be rank deficient, and the singular vectors may not always be perfectly orthogonal. Consequently, it is not uncommon for the matrix undergoing decomposition to fail to meet the criteria for semi-positive definiteness. These circumstances present challenges in generating suitable solutions using these methods. Nonetheless, we believe that the suggestion of employing standard libraries could be beneficial. We are willing to continuous exploration of this potential solution.

---

### Author Response · Authors · 2023-11-21
**Paper Revision**

Dear reviewers,

We greatly appreciate your insightful comments and would like to express our gratitude for your valuable input.

We have taken your feedback seriously and implemented substantial changes to rectify the concerns that were brought up in the reviews.

1. We conduct a comprehensive revision of the abstract, introduction, and main body of the paper. We fix grammatical issues and clarify statements that were presented ambiguously. Considering the title's syntax, we decided to change it to "Fast Updating Truncated SVD for Representation Learning **with** Sparse Matrix." However, due to the constraints of the discussion stage, we cannot make modifications to the title directly. We will also continue to put effort into writing this paper.

2. In the updated version of the paper, we explain certain theoretical definitions in more detail, as suggested. We add the definition of an augmented matrix before the start of the theoretical exposition to make it easier to understand. Additionally, we provide more precise explanations for "approximate" in Theorem 1 and "isometric" in Lemma 3. Notably, we mark all changes in the updated paper in blue, except for grammatical issues and typos.

3. Following the advice of reviewer YLvs, we conducted a retest of the baseline results by utilizing the approach of accessing matrix X through matrix-vector multiplication instead of directly constructing matrix X. As a result, the runtime of the baselines has been reduced in scenarios where the value of 𝜙 is smaller. However, it increases when 𝜙 is large. For both Vecharynski and Yamazaki methods, we have considered the lesser of the two times as the final result and have accordingly updated all experimental findings.

4. We supplement the paper with three more experimental results.

   Runtime of each step. To provide a more comprehensive understanding of how our algorithm optimizes the updating process of the SVD, we conduct additional experiments to measure the runtime of the algorithm for each step of the baseline method. The results are presented in **Appendix E.1**. The results of our experiments show that our approach drastically cuts down on the time needed for both step-1 (QR decomposition) and step-3 (projection processes).

   Efficiency study of synthetic matrices with different sparsities. We investigate the effect of sparsity on the efficiency of updating the SVD using synthetic matrices with varied sparsity levels from 0.01% to 10%. Experimental results in **Appendix E.2** show that the proposed method achieves substantial efficiency improvements for sparse matrices, while the variant using Lanczos vectors performs faster for higher-rank update matrices.

   Effectiveness with varying 𝜙 values.  In **Appendix E.3**, our proposed approach demonstrates improved efficiency while maintaining comparable precision to previous methods, as shown by the Frobenius norm values of the original and reconstructed matrices for different 𝜙 values.

---

> ### Comment · Reviewer_jDYv · 2023-11-22
>
> Perhaps a better title would be "Fast Updating Truncated SVD for Representation Learning with Sparse Matri**ces**"?

---

> > ### Author Response · Authors · 2023-11-23
> >
> > Thank you once again for your valuable feedback. We agree that "Fast Updating Truncated SVD for Representation Learning with Sparse Matrices" is an improved title that accurately reflects the content of our paper.

---

### Meta-Review · Area_Chair_ZrES · 2023-12-07

**Metareview:**

This paper studies the updating truncated Singular Value Decomposition (SVD) for rapidly evolving large-scale data matrices. Existing methods are inefficient because they densify the update matrix and apply the projection to all singular vectors. This paper introduces a novel method that efficiently approximates the truncated SVD of sparse and evolving matrices by leveraging sparsity and storing projections independently. Numerical experiments demonstrate a significant efficiency improvement while maintaining precision compared to previous methods.

The paper received four reviews, with the majority of reviewers commending its novelty and significance. However, two reviewers expressed concerns about the writing quality and the clarity of the presentation. Regarding the numerical stability of the proposed method, one reviewer highlighted concerns, but these were effectively addressed by the authors. While one reviewer commented on the narrowness of the paper's contributions, I believe the overall value justifies its acceptance.

**Justification For Why Not Higher Score:**

The paper suffers from minor issues higlighted by the reviewers.

**Justification For Why Not Lower Score:**

I think the contributions of the paper outweigh its shortcomings.

---

### Decision · Program_Chairs · 2024-01-16

Accept (poster)